# Eye movements reveal spatiotemporal dynamics of visually-informed planning in navigation

**Seren Zhu[1]\*[†], Kaushik J Lakshminarasimhan[2†], Nastaran Arfaei[3], Dora E Angelaki[1,4]**

[1]Center for Neural Science, New York University, New York, United States; [2]Center for Theoretical Neuroscience, Columbia University, New York, United States; [3]Department of Psychology, New York University, New York, United States; [4]Department of Mechanical and Aerospace Engineering, New York University, New York, United States

**Abstract** Goal-oriented navigation is widely understood to depend upon internal maps. Although this may be the case in many settings, humans tend to rely on vision in complex, unfamiliar environments. To study the nature of gaze during visually-guided navigation, we tasked humans to navigate to transiently visible goals in virtual mazes of varying levels of difficulty, observing that they took near-optimal trajectories in all arenas. By analyzing participants' eye movements, we gained insights into how they performed visually-informed planning. The spatial distribution of gaze revealed that environmental complexity mediated a striking trade-off in the extent to which attention was directed towards two complimentary aspects of the world model: the reward location and task-relevant transitions. The temporal evolution of gaze revealed rapid, sequential prospection of the future path, evocative of neural replay. These findings suggest that the spatiotemporal characteristics of gaze during navigation are significantly shaped by the unique cognitive computations underlying real-world, sequential decision making.

**\*For correspondence:**
lt1686@nyu.edu

[†]These authors contributed equally to this work

**Competing interest:** The authors declare that no competing interests exist.

## Editor's evaluation

This beautiful piece of work demonstrates the power of eye movement analysis in understanding the cognitive algorithms for navigation, and more generally, for real-time planning and decision making. Its sophisticated computational measures of the multiple dimensions of eye movement data can potentially inspire discoveries in many fields of cognitive neuroscience concerning rich human behavior.

## Introduction

Planning, the evaluation of prospective future actions using a model of the environment, plays a critical role in sequential decision making (*Hunt et al., 2021*; *Mattar and Lengyel, 2022*). Two-step choice tasks have revealed quantitative evidence that humans are capable of flexible planning (*Momennejad et al., 2017*; *Miller and Venditto, 2021*; *Wunderlich et al., 2012*). Under unfamiliar or uncertain task conditions, planning may depend upon and occur in conjunction with active sensing, the cognitively motivated process of gathering information from the environment (*Kaplan and Friston, 2018*). After all, one cannot make decisions about the future without knowing what options are available. Humans and animals perform near-optimal active sensing via eye movements during binary decision making tasks (*Yang et al., 2016*; *Renninger et al., 2007*) and visual search tasks (*Najemnik and Geisler, 2005*; *Ma et al., 2011*; *Hoppe and Rothkopf, 2019*). Such tasks are typically characterized

by an observation model, a mapping between states and observations, and visual information serves to reduce uncertainty about the state. In contrast, sequential decision-making tasks require knowledge about the structure of the environment characterized by state transitions, and visual information can additionally contribute to reducing uncertainty about this structure. Therefore, the principles of visually-informed decision making uncovered in simplified, discrete settings may not generalize to natural behaviors like real-world navigation, which entails planning a sequence of actions rather than a binary choice. How can we study visually-informed planning in structured, naturalistic sequential decision-making ventures such as navigation?

Theoretical work suggests that information acquisition and navigational planning can be simultaneously achieved through active inference – orienting the sensory apparati to reduce uncertainty about task variables in the service of decision making (*Kaplan and Friston, 2018*). Humans are fortuitously equipped with a highly evolved visual system to perform goal-oriented inference. By swiftly parsing a large, complex scene on a millisecond timescale, the eyes actively interrogate and efficiently gather information from different regions of space to facilitate complex computations (*Leigh and Kennard, 2004*; *Schroeder et al., 2010*). At the same time, eye movements are influenced by the contents of internal deliberation and the prioritization of goals in real-time, providing a faithful readout of important cognitive variables (*Yang et al., 2018*; *Gottlieb and Oudeyer, 2018*; *Hutton, 2008*; *Ryan and Shen, 2020*). Thus, eye tracking lends itself as a valuable tool for investigating how humans and animals gather information to plan action trajectories (*Hoppe et al., 2018*; *Henderson et al., 2013*; *Eckstein et al., 2017*; *Yang et al., 2018*).

Over the past few decades, research on eye movements has led to a growing consensus that the oculomotor system has evolved to prioritize top-down, cognitive guidance over image salience (*Henderson and Hayes, 2017*; *Hayhoe and Ballard, 2005*; *Schütt et al., 2019*). During routine activities such as making tea, we tend to foveate specifically upon objects relevant to the task being performed (e.g. boiling water) while ignoring salient distractors (*Hayhoe and Ballard, 2005*; *Kowler, 2011*). A current consensus about active sensing is that gaze elucidates how humans mitigate uncertainty in a goal-oriented manner (*Yang et al., 2018*). Therefore, we hypothesize that in the context of navigation, gaze will be directed towards the most informative regions of space, depending upon the specific relationship between the participant's position and their goal. A candidate framework to formalize this hypothesis is reinforcement learning (RL), whereby the goal of behavior is cast in terms of maximizing total long-term reward (*Sutton and Barto, 2018*). For example, this framework has been previously used to provide a principled account of why neuronal responses in the hippocampal formation depend upon behavioral policies and environmental geometries (*Gustafson and Daw, 2011*; *Stachenfeld et al., 2017*), as well as a unifying account of how the hippocampus samples memories to replay (*Mattar and Daw, 2018*). Incidentally, RL provides a formal interpretation of active sensing, which can be understood as optimizing information sampling for the purpose of improving knowledge about the environment, allowing for better planning and ultimately greater long-term reward (*Yang et al., 2018*). Here, we invoke the RL framework and hypothesize that eye movements should be directed towards spatial locations where small changes in the local structure of the environment can drastically alter the expected reward.

While active sensing manifests as overt behavior, the planning algorithms which underlie action selection are thought to be more covert (*Hunt et al., 2021*). Researchers have proposed that certain neural codes, such as hierarchical representations, would support efficient navigational planning by exploiting structural redundancies in the environment (*Tomov et al., 2020*; *Solway et al., 2014*). There is also evidence for predictive sequential neural activations during sequence learning and visual motion viewing tasks (*Liu et al., 2019*; *Ekman et al., 2017*; *Kurth-Nelson et al., 2016*). This is reminiscent of replay, a well-documented phenomenon in rodents during navigation (*Johnson and Redish, 2007*; *Pfeiffer and Foster, 2013*; *Dragoi and Tonegawa, 2011*; *Brown et al., 2016*). The mechanism by which humans perform visually-informed planning may similarly involve simulating sequences of actions that chart out potential trajectories, and chunking a chosen trajectory into subgoals for efficient implementation. As recent evidence shows that eye movements reflect the dynamics of internal beliefs during sensorimotor tasks (*Lakshminarasimhan et al., 2020*), we hypothesize that participants' gaze dynamics would also reveal sequential trajectory simulation, and thus reveal the strategies by which humans plan during navigation.

To test both hypotheses mentioned above, we designed a virtual reality navigation task where participants were asked to navigate to transiently visible targets using a joystick in unfamiliar arenas of varying degrees of complexity. We found that human participants balanced foveating the hidden reward location with viewing highly task-consequential regions of space both prior to and during active navigation, and that environmental complexity mediated a trade-off between the two modes of information sampling. The experiment also revealed that participants' eyes indeed rapidly traced the trajectories which they subsequently embarked upon, with such sweeps being more prevalent in complex environments. Furthermore, participants seemed to decompose convoluted trajectories by focusing their gaze on one turn at a time until they reached their goal. Taken together, these results suggest that the spatiotemporal dynamics of gaze are significantly shaped by cognitive computations underlying sequential decision making tasks like navigation.

## Results

### Humans use vision to efficiently navigate to hidden goals in virtual arenas

To study human eye movements during naturalistic navigation, we designed a virtual reality (VR) task in which participants navigated to hidden goals in hexagonal arenas. As we desired to elicit the most naturally occurring eye movements, we used a head-mounted VR system with a built-in eye tracker to provide a full immersion navigation experience with few artificial constraints. Participants freely rotated in a swivel chair and used an analog joystick to control their forward and backward motion along the direction in which they were facing (*Figure 1A*). The environment was viewed from a first-person perspective through an HTC Vive Pro headset with a wide field of view, and several eye movement parameters were recorded using built-in software.

Facilitating quantitative analyses, we designed arenas with a hidden underlying triangular tessellation, where each triangular unit (covering 0.67% of the total area) constituted a *state* in a discrete state space (*Figure 1—figure supplement 1A*). A fraction of the edges of the tessellation was chosen to be impassable barriers, defined as obstacles. Participants could take *actions* to achieve *transitions* between adjacent states which were not separated by obstacles. As participants were free to rotate and/or translate, the space of possible actions was continuous such that participants did not report knowledge about the tessellation. Furthermore, participants experienced a relatively high vantage point and were able to gaze over the tops of all of the obstacles (*Figure 1A*).

On each trial, participants were tasked to collect a *reward* by navigating to a random goal location drawn uniformly from all states in the arena. The goal was a realistic banana which the participants had to locate and foveate in order to unlock the joystick. The banana disappeared 200 ms after foveation, as we wanted to discourage beaconing. Participants were instructed to press a button when they believed that they have arrived at the remembered goal location. Then, feedback was immediately displayed on the screen, showing participants that they had received either two points for stopping within the goal state, one point for stopping in a state sharing a border with the goal state (up to three possible), or zero for stopping in any other state. While participants viewed the feedback, a new goal for the next trial was spawned without breaking the continuity of the task. In separate blocks, participants navigated to 50 goals in each of five different arenas (*Figure 1B*). All five arenas were designed by defining the obstacle configurations such that the arenas varied in the average path length between two states, as quantified by the average state closeness centrality (Methods – *Equation 2*; *Figure 1—figure supplement 1B*, *Javadi et al., 2017*). One of the blocks involved an entirely open arena that contained only a few obstacles at the perimeter, such that on most trials, participants could travel in a straight line to the goal location (*Figure 1B* – leftmost). On the other extreme was a maze arena in which most pairs of states were connected by only one viable path (*Figure 1B* – rightmost). Because lower centrality values correspond to more complex arenas, negative centrality can be interpreted as a measure of arena complexity. For simplicity, we defined complexity using a linear transformation of negative centrality such that the open arena had a complexity value of zero, and we used this scale throughout the paper (see Methods; *Figure 1C*). We captured both within-participant and between-participant variability by fitting linear mixed effects (LME) models with random slopes and intercepts to predict trial-specific outcomes, and found consistent effects across participants.

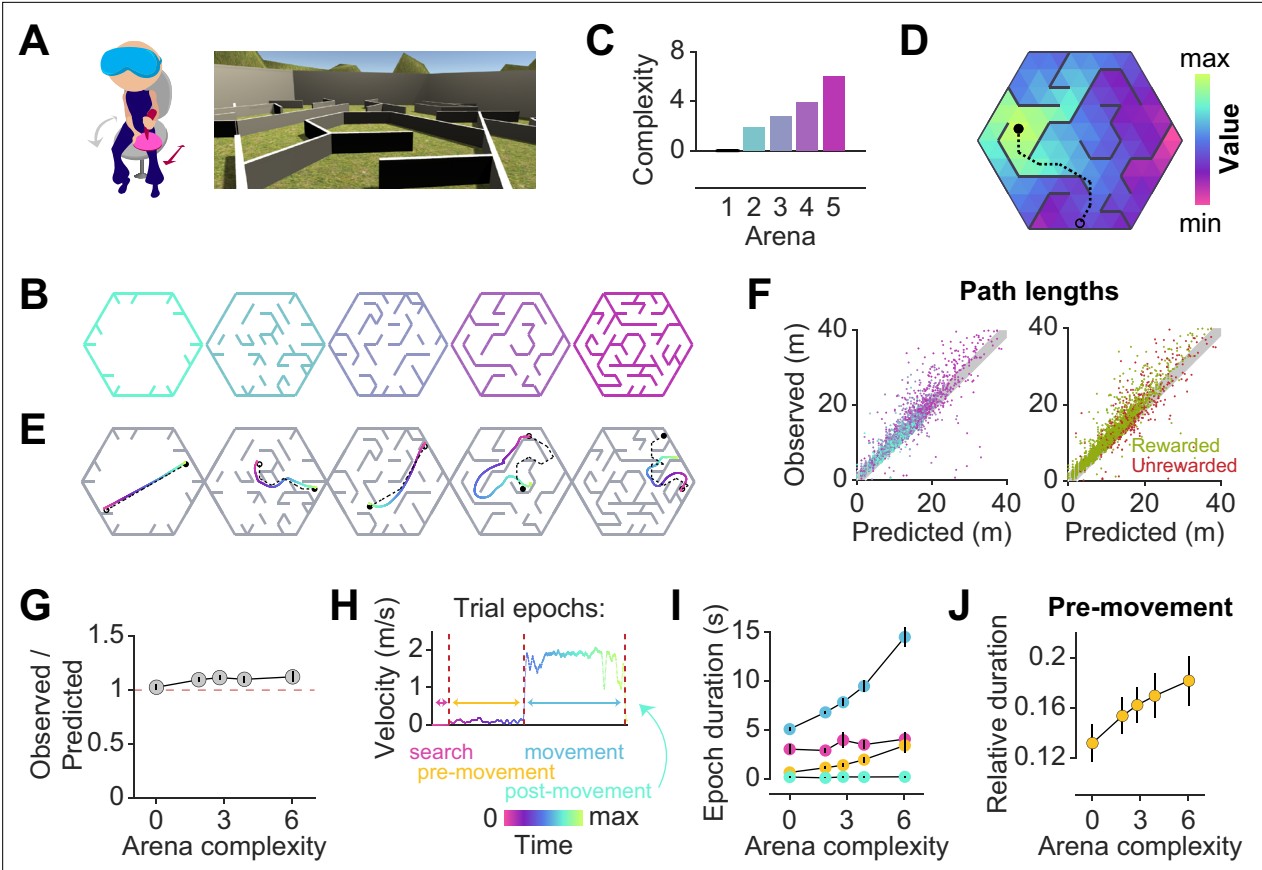

**Figure 1.** Participants exhibit near-optimal navigation performance across multiple environments. (**A**) Left: Human participants wore a VR headset and executed turns by rotating in a swivel chair, while translating forwards or backwards using an analog joystick. Right: A screenshot of the first-person view of the display. The headset conferred an immersive field of view of 110°. (**B**) Aerial view showing the layout of the arenas. (**C**) Arenas ranged in complexity, which is related to negative mean state closeness centrality. (**D**) Heatmap showing the value function corresponding to an arbitrary goal state (closed circle) in one of the arenas. The value of each state is related to the geodesic distance between that state and the goal. Dashed line denotes the optimal trajectory from an example starting state (open circle). (**E**) Trajectories from an example trial in each arena, executed by one participant. The optimal trajectory is superimposed in black (dashed line). Time is color-coded. (**F**) Comparison of the empirical path length against the path length predicted by the optimal policy. The gray shaded region denotes the width of the outer reward zone (see *Figure 1—figure supplement 1A*). Left: Data points are colored in accordance to the colors of each arena as depicted in B. Right: Unrewarded trials (red) vs rewarded trials (green) had similar path lengths. For both plots, all trials for all participants and all arenas are superimposed. (**G**) Across participants, the average ratio of observed vs optimal (predicted) trajectory lengths is consistently around 1 in all arenas. (**H**) The search epoch was defined as the period between goal stimulus appearance and goal stimulus foveation. A threshold applied on the filtered joystick input (movement velocity) was used to delineate the pre-movement and movement epochs. (**I**) The average duration of the pre-movement (orange) and movement epochs (blue; colored according to the scheme in H) increased with arena complexity, in conjunction with the trial-level effects exerted by path lengths (*Figure 1—figure supplement 3A*). (**J**) The relative planning time, calculated as the ratio of pre-movement to total trial time after goal foveation, was higher for more complex arenas. For G, I, and J, error bars denote ±1 SEM.

The online version of this article includes the following figure supplement(s) for figure 1:

**Figure supplement 1.** Experimental details and behavioral performance.

**Figure supplement 2.** Effect of arena complexity on behavioral performance.

**Figure supplement 3.** Effect of path length and arena complexity on epoch durations and gaze.

---

Therefore, we primarily show average trends in the main text, but participant-specific effects are included in *Appendix 2—Tables 1–4*.

To quantify behavioral performance, we first computed the optimal trajectory for each trial using dynamic programming, an efficient algorithm with guaranteed convergence. This technique uses two pieces of information — the goal location (*reward function*) and the obstacle configuration (*transition structure*) — to find an optimal *value function* over all states such that the value of each state is

equal to the (negative) length of the shortest path between that state and the goal state (*Figure 1D*, *Figure 1—figure supplement 1C*). The *optimal policy* requires that participants select actions to climb the value function along the direction of steepest ascent, which would naturally bring them to the goal state while minimizing the total distance traveled. *Figure 1E* shows optimal (dashed) as well as behavioral (colored) trajectories from an example trial in each arena. Behavioral path lengths were computed by integrating changes in the participants' position in each trial.

Although participants occasionally took a suboptimal route (*Figure 1E* – second from right), they took near-optimal paths (i.e. optimal to within the width of the reward zone) on most trials (*Figure 1F*), scoring (mean±SD across participants) 72 ± 7% of the points across all arenas and stopping within the reward zone on 85 ± 6% of all trials (*Figure 1—figure supplement 1D–E*). We quantified the degree of optimality by computing the ratio of observed vs optimal path lengths to the participants' stopping location. Across all rewarded trials, this ratio was close to unity (1.1 ± 0.1), suggesting that participants were able to navigate efficiently in all arenas (*Figure 1G*). Navigational performance was near-optimal from the beginning, such that there was no visible improvement with experience (*Figure 1—figure supplement 2A*). Even on unrewarded trials, participants took a trajectory that is, on average, only 1.2±0.1 times the optimal path length from the participant's initial state to their stopping location (*Figure 1E* – rightmost, *Figure 1—figure supplement 2B*). This suggests that remembering the goal location was not straightforward. In fact, the fraction of rewarded trials decreased with increasing arena complexity (Pearson's r(63) = –0.64, p=$8 \times 10^{-9}$), suggesting that the ability to remember the goal location is compromised in challenging environments (*Figure 1—figure supplement 2C*). Each trial poses unique challenges for the participant, such as the number of turns in the trajectory, the length of the trajectory, and the angle between the initial direction of heading and the direction of target approach (relative bearing). Among these variables, the length of the trajectory best predicts the error in the participants' stopping position (*Figure 1—figure supplement 1F*).

In order to understand how participants tackled the computational demands of the task, it is critical to break down each trial into three main epochs: *search* – when participants sought to locate the goal, *pre-movement* – when participants surveyed their route prior to utilizing the joystick, and *movement* – when participants actively navigated to the remembered goal location (*Figure 1H*). On some trials, participants did not end the trial via button press immediately after stopping, but this post-movement period constituted a negligible proportion of the total trial time. Although participants spent a major portion of each trial navigating to the target, the relative duration of other epochs was not negligible (mean fraction ± SD – search: 0.27 ± 0.05, pre-movement: 0.11 ± 0.03, movement: 0.60 ± 0.06; *Figure 1—figure supplement 2D*). There was considerable variability across participants in the fraction of time spent in the pre-movement phase (coefficient of variation (CV) – search: 0.18, pre-movement: 0.31, movement: 0.10), although this did not translate to a significant difference in navigational precision (*Figure 1—figure supplement 2E*). One possible explanation is that some participants were simply more efficient planners or were more skilled at planning on the move. While the duration of the search epoch was similar across arenas, the movement epoch duration increased drastically with increasing arena complexity (*Figure 1I*). This was understandable as the more complex arenas posed, on average, longer trajectories and more winding paths by virtue of their lower centrality values. Notably, the pre-movement duration was also higher in more complex arenas, reflecting the participant's commitment to meet the increased planning demands in those arenas (*Figure 1J*). Nonetheless, the relative pre-movement duration was similar for rewarded and unrewarded trials (*Figure 1—figure supplement 2F*). This suggests that the participants' performance is limited by their success in remembering the reward location, rather than in meeting planning demands. On a finer scale, the duration of the pre-movement and movement epochs were both strongly influenced by the path length and the number of turns, but not by the bearing angle (*Figure 1—figure supplement 2G*). Overall, these results suggest that in the presence of unambiguous visual information, humans are capable of adapting their behavior to efficiently solve navigation problems in complex, unfamiliar environments.

## A computational analysis supports that human eye movements are task relevant

Aiming to gain insights from participants' eye movements, we begin by examining the spatial distribution of gaze positions during different trial epochs (*Figure 2A*). Within each trial, the spatial spread of

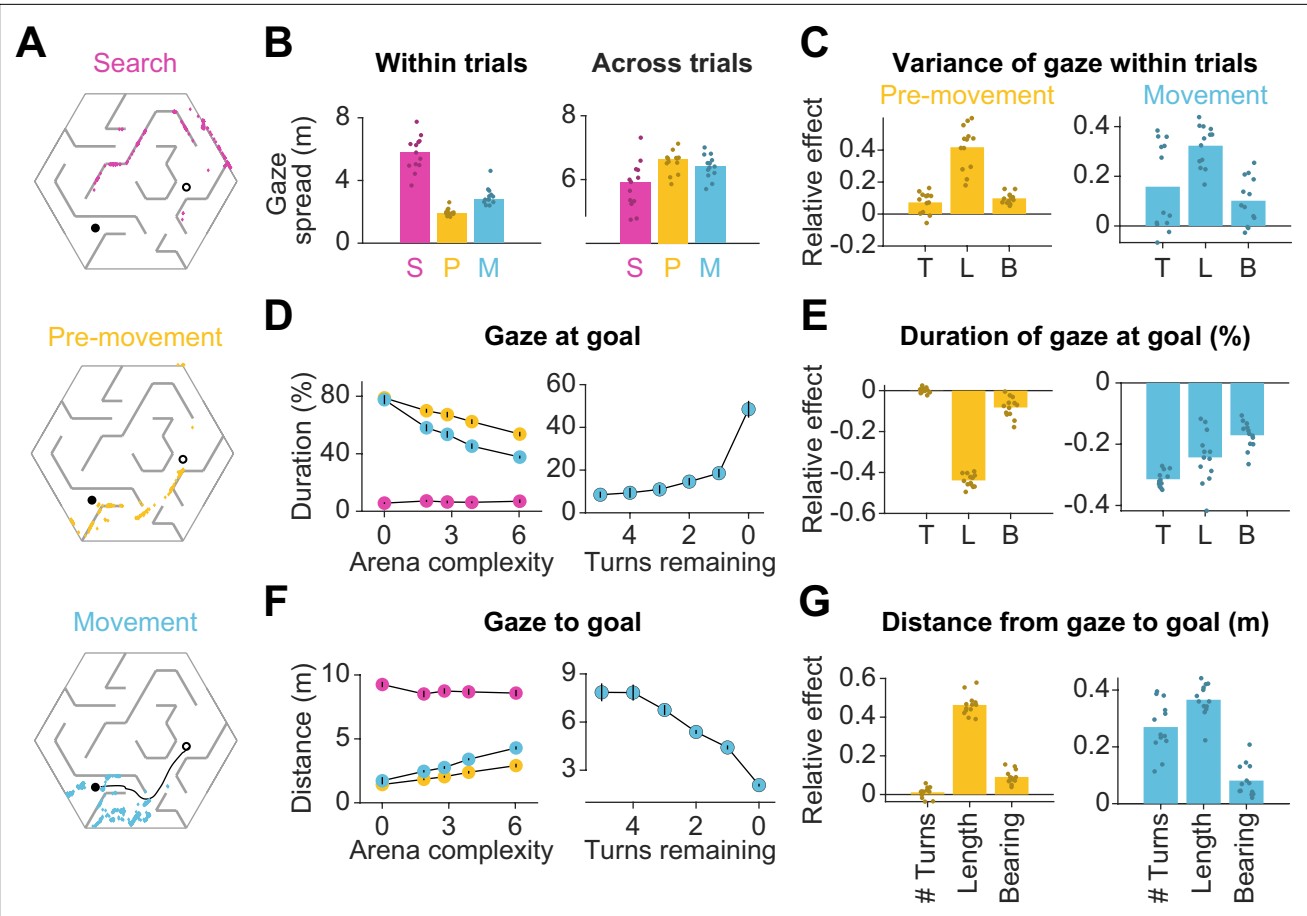

**Figure 2.** Eye movements are modulated by goal location and environment complexity. (**A**) Eye positions on a representative trial for one participant during the three main trial epochs. Each datapoint corresponds to one frame. An open black circle denotes the start location, while a closed black circle denotes the goal. The color scheme applies to all plots in this figure. (**B**) Left: The median spatial spread of gaze within trial epochs (averaged across trials and arenas) was higher during search than during pre-movement and movement. Right: In contrast, the median spread of the average gaze positions across trials was higher during the pre-movement and movement epochs. Individual participant data are overlaid on top of the bars. (**C**) Left: A linear mixed model for the effect of trial-specific variables (number of turns, length of optimal trajectory, relative bearing) on the variance of gaze within the pre-movement epoch reveals that the expected path length has the greatest effect on gaze spread. The overlaid scatter shows fixed effect slope + participant - specific random effect slope. Right: Similar result for gaze spread within the movement epoch. (**D**) Left: Across participants, the average fraction of time for which gaze was near (within 2 m of) the center of the goal state decreased with arena complexity. The arena-level variable (complexity) and the trial-level equivalent (path length) both independently exert effects on the amount of time subjects looked at the goal (*Figure 1— figure supplement 3B*). Right: Participants spent more time looking near the goal location when fewer turns separated them from the goal. (**E**) Left: A linear mixed model reveals that expected path length had the greatest negative effect on the fraction of time that participants spent gazing at the goal location prior to movement. Right: During movement, all measures of trial difficulty decreased goal-fixation behavior, especially the number of turns. (**F**) Left: The average distance between the gaze position and the goal state increased with arena complexity during pre-movement and movement. Right: The average distance of the point of gaze from the goal location decreases as the participant approaches the target. (**G**) Left: Expected path length best predicted the average distance of gaze to the goal prior to movement. Right: During movement, the number of turns and the expected path length most positively affected this statistic. All error bars denote ±1 SEM, and all variables were z-scored prior to model fitting.

The online version of this article includes the following figure supplement(s) for figure 2:

**Figure supplement 1.** Gaze variance.

**Figure supplement 2.** Gaze locations.

the gaze position was much larger during visual search than during the other epochs (mean spread ± SD across participants – search: 5.8 ± 1.1 m, pre-movement: 2.0 ± 0.2 m, movement: 3.0 ± 0.6 m; *Figure 2B* – left). This pattern was reversed when examining the spatial spread across trials (mean spread ± SD – search: 5.8 ± 0.7 m, pre-movement: 6.5 ± 0.3 m, movement: 6.4 ± 0.4; *Figure 2B* – right). This suggests that participants' eye movements during pre-movement and movement were

chiefly dictated by trial-to-trial fluctuations in task demands. Furthermore, the variance of gaze positions within a trial, both prior to and during movement, was largely driven by the path length (*Figure 2C*) and therefore increased with arena complexity (*Figure 2—figure supplement 1A*).

How did the task demands constrain human eye movements? Studies have shown that reward circuitry tends to orient the eyes toward the most valuable locations in space (*Hikosaka et al., 2006*; *Koenig et al., 2017*). Moreover, when the goal is hidden, it has been argued that fixating the hidden reward zone may allow for the oculomotor circuitry to carry the burden of remembering the latent goal location (*Lakshminarasimhan et al., 2020*; *Postle et al., 2006*). Consistent with this, participants spent a large fraction of time looking at the reward zone, and this statistic was interestingly higher during pre-movement (66 ± 10%) than during movement (54 ± 6%). However, goal fixation decreased with arena complexity (*Figure 2D* – left, *Figure 2—figure supplement 2A* – left), resulting in a larger mean distance between the gaze and the goal in more complex arenas (*Figure 2F* – left, *Figure 2—figure supplement 2A* – right). This effect could not be attributed to participants forgetting the goal location in more complex arenas, as we found a similar trend when analyzing gaze in relation to the eventual stopping location (which could be different from the goal location; *Figure 2—figure supplement 1B*). A more plausible explanation is that looking solely at the goal might prevent participants from efficiently learning the task-relevant transition structure of the environment, as the structure is both more instrumental to solving the task and harder to comprehend in more challenging arenas. If central vision is attracted to the remembered goal location only when planning demands are low, this tendency should become more prevalent as participants approach the target. Indeed, participants spend significantly more time looking at the goal when there is a straight path to the goal than when the obstacle configuration requires that they make at least one turn prior to arriving upon such a straight path (*Figure 2D and F* – right). Also in alignment with this explanation, trial-level analyses revealed that during pre-movement, the tendency to look at the goal substantially decreased with greater path lengths (*Figure 2E and G* – left). During movement, goal fixation has more diverse influences from affordances linked to navigation, especially as a greater amount of turns also decreased the amount of time participants dedicated to looking at the remembered goal location (*Figure 2E and G* – right).

As mentioned earlier, computing the optimal trajectory requires precisely knowing both the reward function as well as the transition structure. While examining the proximity of gaze to the goal reveals the extent to which eye movements are dedicated to encoding the reward function, how may we assess the effectiveness with which participants interrogate the transition structure of the environment to solve the task of navigating from point A to point B? In the case that a participant has a precise model of the transition structure of the environment, they would theoretically be capable of planning trajectories to the remembered goal location without vision. However, in this experiment, the arena configurations were unfamiliar to the participants, such that they would be quite uncertain about the transition structure. The finding that participants achieved near-optimal performance on even the first few trials in each arena (*Figure 1—figure supplement 2A*) indicates that humans are capable of using vision to rapidly reduce their uncertainty about the aspects of the model needed to solve the task. This reduction in uncertainty could be accomplished in two ways: (i) by actively sampling visual information about the structure of the arena in the first few trials and then relying largely on the internal model later on after this information is consolidated, or (ii) by actively gathering visual samples throughout the experiment on the basis of immediate task demands on each trial. We found evidence in support of the second possibility — arena-specific pre-movement epoch durations did not significantly decrease across trials (*Figure 2—figure supplement 2B*).

Did participants look at the most informative locations? Depending on the goal location, misremembering the location of certain obstacles would have a greater effect on the subjective value of actions than for other obstacles (*Appendix 1—figure 1A*, "Relevance simulations" in Appendix 1). We leveraged this insight and defined a metric to quantify the task-*relevance* of each transition by computing the magnitude of the change in value of the participant's current state, for a given goal location, if the status of that transition was misremembered:

$$\Omega_k(s_0, s_G) = [V(s_0|T_k = 1) - V(s_0|T_k = 0)]^2 \tag{1}$$

where $\Omega_k(s_0, s_G)$ denotes the relevance of the $k^{\text{th}}$ transition for navigating from state $s_0$ to the goal state $s_G$, $T_k$ denotes the status of that transition (1 if it is passable and 0 if it is an obstacle), and

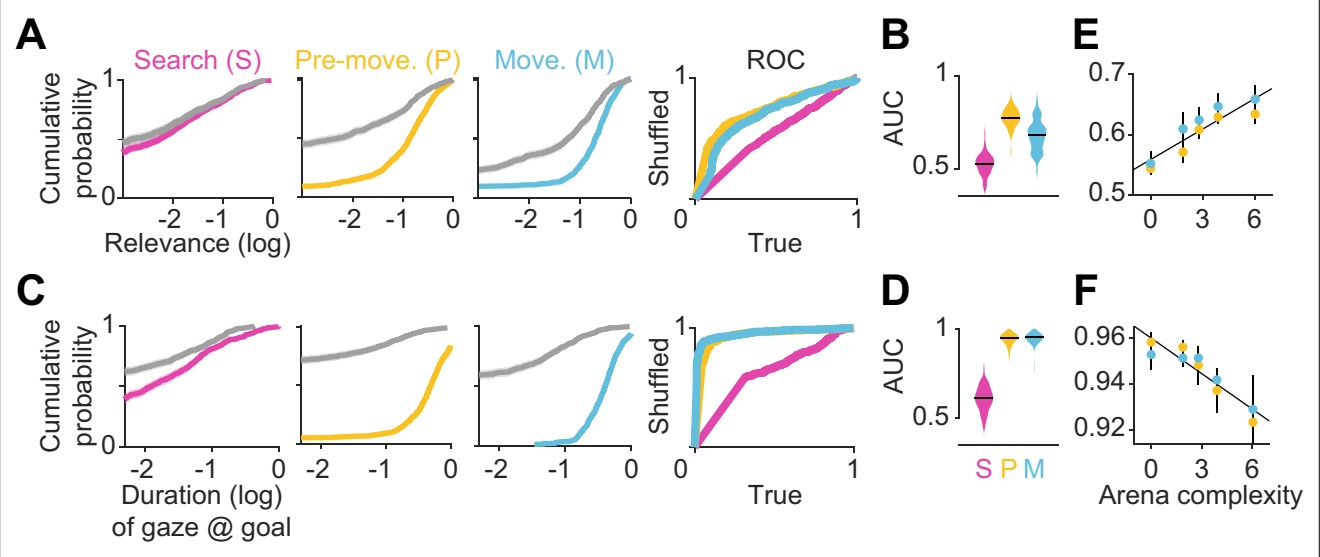

**Figure 3.** Eye movements reveal a cognitive trade-off between reward and transition encoding. (**A**) Left: Cumulative distribution (computed by pooling trials from all participants) of average log (normalized) relevance values (colored line) and the corresponding shuffled distribution (gray) during search (left), pre-movement (center), and movement (right) epochs (data for the most complex arena is shown). Shaded regions denote 95% confidence bounds computed using Greenwood's formula. Rightmost: ROC curves characterizing the gaze relevance during the three epochs. (**B**) Area under the ROC curves (AUC) for different epochs, colored according to the color scheme in A. (**C-D**) Similar plots as A-B, but for the distributions of the log fraction of the duration in each epoch spent gazing near (within 2 m of) the eventual stopping position (which was assumed to be the participants' believed goal location). (**E**) AUC values of gaze relevance computed for the distributions of trial-averaged relevances, after excluding fixations within the reward zone, during the pre-movement (orange) and movement (blue) epochs. Black line represents best-fit linear regression model. (**F**) Similar to E, but showing the AUC values of gaze durations within the reward zone. All error bars were computed using bootstrapping.

The online version of this article includes the following figure supplement(s) for figure 3:

**Figure supplement 1.** Gaze relevance across epochs and arenas.

$V(s_0|T_k = 1)$ denotes the value of state $s_0$ computed with respect to the goal state $s_G$ by setting $T_k$ to 1. It turns out that this measure of relevance is directly related to the magnitude of expected change in subjective value of the current state when looking at the $k^{\text{th}}$ transition, provided that the transitions are stationary and the participant's uncertainty is uniform across transitions (Supplementary Notes). Thus, maximally relevant transitions identified by *Equation 1* are precisely those which may engender the greatest changes of mind about the current return (i.e. utility) of the current state. On each trial, the transitions with the highest relevance strikingly correspond to bottleneck transitions that bridge clusters of interconnected states (*Appendix 1—figure 1B*). Relevance was also high for obstacles that precluded a straight path to the goal, as well as for transitions along the optimal trajectory (*Appendix 1—figure 1C*). By defining relevance of transitions according to *Equation 1*, we can thus capture multiple task-relevant attributes in a succinct manner. In Appendix 1, we point to a generalization of this relevance measure for settings in which the transition structure is stochastic (e.g. in volatile environments) and the subjective uncertainty is heterogeneous (i.e. the participant is more certain about some transitions than others).

We quantified the usefulness of participants' eye position on each frame as the relevance of the transition closest to the point of gaze, normalized by that of the most relevant transition in the entire arena given the goal state on that trial. Then, we constructed a distribution of shuffled relevance values by analyzing gaze with respect to a random goal location. *Figure 3A* shows the resulting cumulative distributions across trials for the average participant during the three epochs in an example arena. As expected, the relevance of participants' gaze was not significantly different from chance during the search epoch, as the participant had not yet determined the goal location. However, relevance values were significantly greater than chance both during pre-movement and movement (median relevance for the most complex arena, pre-movement – true: 0.14, shuffled: 0.006; movement – true: 0.20, shuffled: 0.06; see *Figure 3—figure supplement 1*, *Appendix 2—table 2* for other arenas).

To concisely describe participants' tendency to orient their gaze toward relevant transitions in a scale-free manner, we constructed receiver operating characteristic (ROC) curves by plotting the cumulative probability of shuffled gaze relevances against the cumulative probability of true relevances (*Figure 3A* – rightmost). An area under the ROC curve (AUC) greater (less) than 0.5 would indicate that the gaze relevance was significantly above (below) what is expected from a random gaze strategy. Across all arenas, the AUC was highest during the pre-movement epoch (*Figure 3B*; mean AUC ± SD – search: 0.52 ± 0.03, pre-movement: 0.77 ± 0.03, movement: 0.68 ± 0.07). This suggests that participants were most likely to attend to relevant transitions when contemplating potential actions before embarking upon the trajectory.

As the most relevant transitions can sometimes be found near the goal (e.g. *Figure 1B* – left), we investigated whether our evaluation of gaze relevance was confounded by the observation that participants spent a considerable amount of time looking at the goal location (*Figure 2C*). Therefore, we first quantified the tendency to look at the goal location in a manner analogous to the analysis of gaze relevance (*Figure 3A–B*) by computing the AUC constructed using the true vs shuffled distributions of the duration spent foveating the goal in each epoch (*Figure 3C*). Across all arenas, AUCs were high during the pre-movement and movement epochs, confirming that there was a strong tendency for participants to look at the goal location (*Figure 3D*). When we excluded gaze positions that fell within the reward zone while computing relevance, we found that the degree to which participants looked at task-relevant transitions outside of the reward zone increased with arena complexity: the tendency to look at relevant transitions was greater in more complex arenas, falling to chance for the easiest arena (*Figure 3E*; Pearson's r(63) – pre-movement: 0.40, p=0.004; movement: 0.28, p=0.05). In contrast, the tendency to look at the goal location followed the opposite trend and was greater in easier arenas (*Figure 3F*; Pearson's r(63) – pre-movement: –0.46, p=0.0003; movement: –0.33, p=0.001). These analyses reveal a striking trade-off in the allocation of gaze between encoding the reward function and transition structure that closely mirrors the cognitive requirements of the task. This trade-off is not simply a consequence of directing gaze more/less often at the reward, as such temporal statistics are preserved while shuffling. Instead, it points to a strategy of directing attention away from the reward and towards *task-relevant* transitions in complex arenas. This compromise allowed participants to dedicate more time to surveying the task-relevant structure in complex environments and likely underlies their ability to take near-optimal paths in all environments, albeit at the cost of an increased tendency to forget the precise goal location in complex environments (*Figure 1—figure supplement 2C*). The trade-off reported here is roughly analogous to the trade-off between looking ahead towards where you're going and having to pay attention to signposts or traffic lights. One could get away with the former strategy while driving on rural highways whereas city streets would warrant paying attention to many other aspects of the environment to get to the destination.

## The temporal evolution of gaze includes distinct periods of sequential prospection

So far, we have shown that the spatial distribution of eye movements adapts to trial-by-trial fluctuations in task demands induced by changing the goal location and/or the environment. However, planning and executing optimal actions in this task requires dynamic cognitive computations within each trial. To gain insights into this process, we examined the temporal dynamics of gaze. *Figure 4A* shows a participant's gaze in an example trial which has been broken down into nine epochs (pre-movement: I–VI, movement: VII–IX) for illustrative purposes (see *Video 1* for more examples). The participant initially foveated the goal location (epoch I), and their gaze subsequently traced a trajectory *backwards* from the goal state towards their starting position (II) roughly along a path which they subsequently traversed on that trial (dotted line). This sequential gaze pattern was repeated shortly thereafter (IV), interspersed by periods of non-sequential eye movements (III and V). Just before embarking on their trajectory, the gaze traced the trajectory, now in the *forward* direction, until the end of the first turn (VI). Upon reaching the first turning point in their trajectory (VII), they executed a similar pattern of sequential gaze from their current position toward the goal (VIII), tracing out the path which they navigated thereafter (IX). We refer to the sequential eye movements along the future trajectory in the backwards and forwards direction as *backward sweeps* and *forward sweeps*, respectively. During such sweeps, participants seemed to rapidly navigate their future paths with their eyes, and all participants exhibited sweeping eye movements without being explicitly instructed to plan their trajectories prior

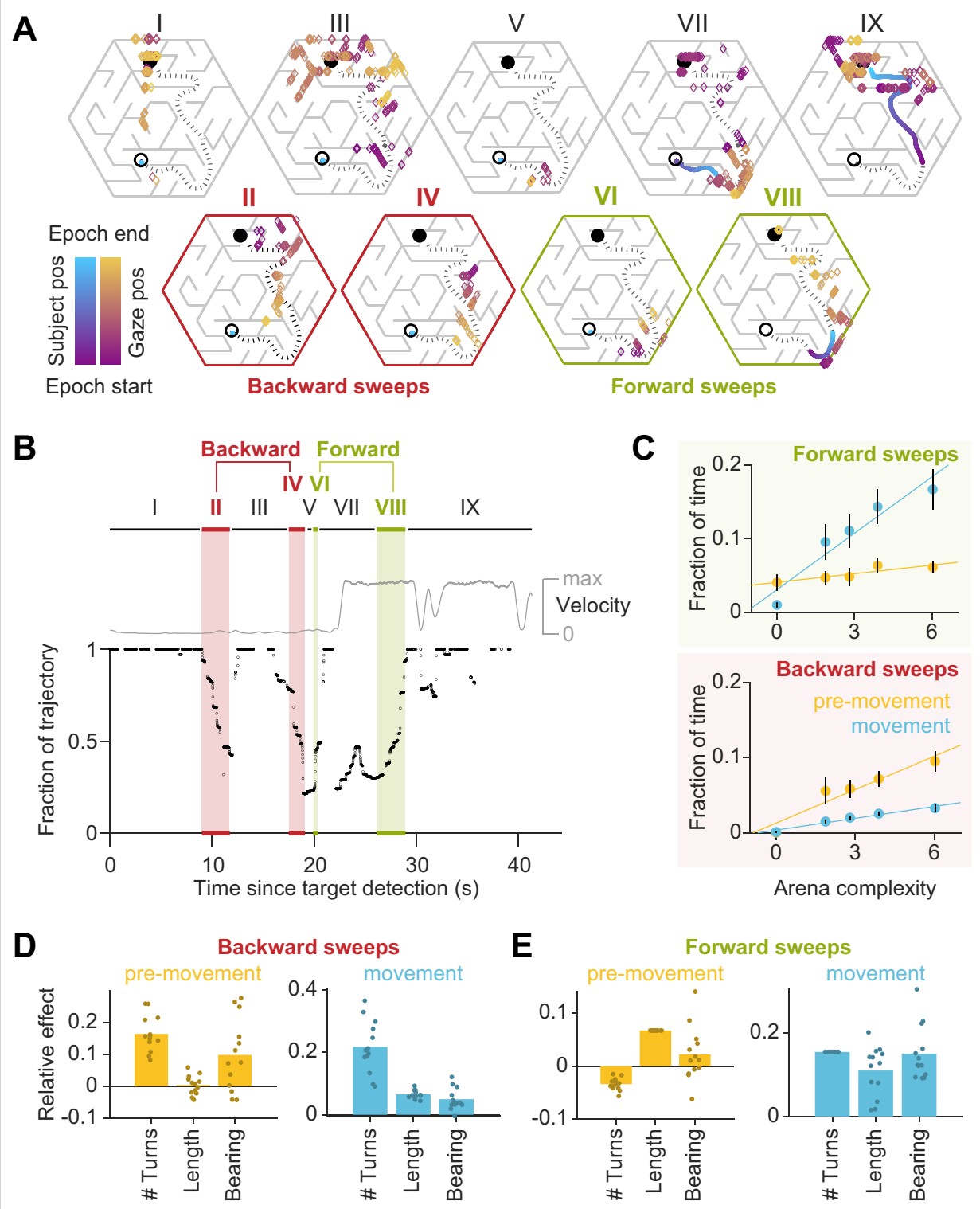

**Figure 4.** Gaze traveled forwards and backwards along the intended trajectory. (**A**) Spatial locations of gaze positions (the arrow of relative time within each window increases from violet to orange) and participant positions (violet to blue) during individual time windows demarcated in panel B. Panels in the bottom row correspond to time periods corresponding to sweeps. The participant's trajectory from the starting location (open black circle) to the goal (closed black circle) is denoted by a black dashed line. (**B**) Time-series of the points on the trajectory that were closest to the participant's gaze on each frame, expressed as a fraction (0: start of trajectory, 1: end of trajectory) during one example trial. Only frames during which the gaze position fell within 2 m of the trajectory are plotted. The gray trace shows the movement velocity of the participant during this trial. Red and green shaded regions highlight time windows during which the sweep classification algorithm detected backward and forward sweeps, respectively. In this trial, there

*Figure 4 continued on next page*

*Figure 4 continued*

were two backward sweeps before movement, and one forward sweep each before and during movement. (**C**) Across all participants, the fraction of time spent sweeping in the forward and backward directions within each epoch reveals an antiparallel effect: more time was spent sweeping forwards during movement than during pre-movement (top), whereas more time was spent sweeping backwards during pre-movement than during movement (bottom). Generally, the arena complexity as well as the trial-specific path lengths, both increase the fraction of time sweeping (*Figure 1—figure supplement 3C*). Error bars denote ±1 SEM. (**D**) Linear mixed models with random intercepts and slopes for the effect of trial-specific variables (number of turns, length of optimal trajectory, relative bearing) on the fraction of time that participants spent sweeping their trajectory in the backward direction, separated for pre-movement and movement epochs. The overlaid scatter shows fixed effect slope + participant – specific random effect slope. (**E**) Similar analysis as **D**, but for forward sweeps. All variables were z-scored prior to model fitting.

The online version of this article includes the following figure supplement(s) for figure 4:

**Figure supplement 1.** Effect of arena complexity and alternative trajectories on gaze.

**Figure supplement 2.** Properties of sweeping eye movements.

**Figure supplement 3.** Sweep direction and timing.

to navigating. The fraction of time that participants looked near the trajectories which they subsequently embarked upon increased with arena and trial difficulty (*Figure 4—figure supplement 1A*). To algorithmically detect periods of sweeps, gaze positions on each trial were projected onto the trajectory taken by the participant by locating the positions along the trajectory closest to the point of gaze on each frame (Methods). On each frame, the length of the trajectory up until the point of the gaze projection was divided by the total trajectory length, and this ratio was defined as the 'fraction of trajectory'. We used the increase/decrease of this variable to determine the start and end times of periods when the gaze traveled sequentially along the trajectory in the forward/backward directions (sweeps) for longer than chance (*Figure 4B*; see Methods).

The environmental structure exerted a strong influence on the probability of sweeping: the fraction of trials in which this phenomenon occurred was significantly correlated with arena complexity (Pearson's $r(63) = 0.73$, $p=5e^{-12}$; *Figure 4—figure supplement 1B*). This suggests that sweeping eye movements could be integral to trajectory planning. Most notably, on average, backward sweeps occupied a greater fraction of time during pre-movement than during movement, but forward sweeps predominantly occurred during movement (backward sweeps – pre-movement: 5.6 ± 3.3%, movement: 1.9 ± 0.6%; forward sweeps – pre-movement: 5.2 ± 3.0%, movement: 10.5 ± 6.3%; *Figure 4C*). This suggests that the initial planning is primarily carried out by sweeping backwards from the goal. Furthermore, trials with a greater number of turns and those in which participants initially move away from the direction of the target tend to have more backward sweeps during pre-movement (*Figure 4D* – left). In contrast, the number of turns inhibited forward sweeps during pre-movement, which were instead driven largely by the length of the trajectory (*Figure 4E* – left). During movement on the other hand, multiple measures of trial difficulty increased the likelihood of forward sweeps (*Figure 4E* – right), whereas backward sweeps depended primarily on the number of turns (*Figure 4D* – right). Together, these dependencies explain why backward sweeps are more common during pre-movement but forward sweeps dominate during movement.

Aside from gazing upon the target or the trajectory on each trial, about 20% of eye movements were made to other locations in space (*Figure 4—figure supplement 1C*). Besides task-relevant locations such as bottleneck transitions, we wanted to know whether these other locations also comprised alternative trajectories to the goal. To test this, we identified all trajectories whose path lengths were comparable (within about 1 SD; Methods) to the chosen trajectory on each trial. The fraction of time spent looking at alternative trajectories and the chosen trajectory both increased with the number of alternatives

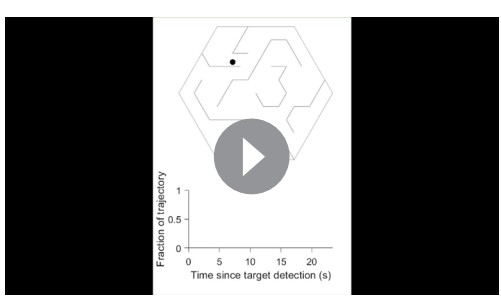

**Video 1.** Six representative trials in which participants exhibited sweeping eye movements. (Top) Aerial view of the arena with the participant's dynamically evolving position (lilac) and gaze (green). The target is represented as a black circle. (Bottom) Time-evolving version of the plot described in Figure 4B. The video speed is veridical, and the search epoch was omitted from each trial.

https://elifesciences.org/articles/73097/figures#video1

(Pearson's r(14) = 0.91, p=1e$^{-6}$ pre-movement; *r*=0.90, p=2e$^{-6}$ movement), suggesting that participants engage in some form of deliberation (*Figure 4—figure supplement 1D*). This deliberation happens at the expense of reducing the fraction of time spent looking at the hidden goal location (r(14) = –0.85, p=3e$^{-5}$ pre-movement; *r*=–0.64, *P*=7e$^{-3}$ movement), revealing that the planning algorithm is likely subject to a cognitive trade-off mediated by the number of available options, in addition to that mediated by model complexity demonstrated earlier.

When looking at the trajectory, the mean speed of backward sweeps was greater than the speed of forward sweeps across all arenas (backward sweeps: 26 ± 4 m/s, forward sweeps: 21 ± 3 m/s; *Figure 4—figure supplement 2A*). Notably, sweep velocities were more than 10 × greater than the average participant velocity during the movement epoch (1.4 ± 0.1 m/s). This is reminiscent of the hippocampal replay of trajectories through space, as such sequential neural events are also known to be compressed in time (around 2–20 × the speed of neural sequence activation during navigation) (*Buhry et al., 2011*). Both sweep speeds and durations slightly increased with arena complexity (*Figure 4—figure supplement 2A*). This is because peripheral vision processing must lead the control of central vision to allow for sequential eye movements to trace a viable path (*Caspi et al., 2004*; *Crowe et al., 2000*). In more complex arenas such as the maze where the search tree is narrow and deep, the obstacle configuration is more structured and presents numerous constraints, and thus path tracing computations might occur more quickly. Accordingly, longer path lengths best predicted sweep speeds (*Figure 4—figure supplement 2B*). However, due to the lengthier and more convoluted trajectories in those arenas, the gaze must cover greater distances and make more turns, resulting in sweeps which last longer (*Figure 4—figure supplement 2A, C*). Another property of sweeps is that they comprised more saccades in more difficult trials (*Figure 4—figure supplement 2A, D*), and saccade rates were higher during sweeps than at other times after goal detection (*Figure 4—figure supplement 2F*). This suggests that either visual processing during sweeps was expedited compared to average, or sweeps resulted from eye movements which followed a pre-planned saccade sequence.

If the first sweep on a trial occurred during pre-movement, the direction of the sweep was more likely to be backwards, while if the first sweep occurred during movement, it was more likely to be in the forwards direction (*Figure 4—figure supplement 3A, B*). The latency between goal detection and the first sweep increased with arena difficulty (*Figure 4—figure supplement 3C* – top row), and more specifically the number of turns and expected path length (*Figure 4—figure supplement 3C* – bottom row), suggesting that sweep initiation is preceded by brief processing of the arena, and more complex tasks elicited longer processing. While the sequential nature of eye movements could constitute a swift and efficient way to perform instrumental sampling, we found that task-relevant eye movements were not necessarily sequential. When we reanalyzed the spatial distribution of gaze positions by removing periods of sweeping, the resulting relevance values remained greater than chance (*Figure 4—figure supplement 3D*).

What task conditions promote sequential eye movements? To find out, we computed the probability that the participants engaged in sweeping behavior as a function of time and position, during the pre-movement and movement epochs respectively, and focusing on the predominant type of sweep during those periods (backward and forward sweeps respectively; *Figure 4C*). During pre-movement, we found that the probability of sweeping gradually increased over time, suggesting that backward sweeps during the initial stages of planning are separated from the time of target foveation by a brief pause, during which participants may be gathering some preliminary information about the environment (*Figure 5A* – left). During movement, on the other hand, the probability of sweeping is strongly influenced by whether participants are executing a turn in their trajectory. Obstacles often preclude a straight path to the remembered goal location, and thus participants typically find themselves making multiple turns while actively navigating. Consequently, a trajectory may be divided into a series of straight segments separated by brief periods of elevated angular velocity. We isolated such periods by applying a threshold on angular velocity, designating the periods of turns as *subgoals*, and aligned the participant's position in all trials with respect to the subgoals. The likelihood of sweeping the trajectory in the forward direction tended to spike precisely when participants reached a subgoal (*Figure 5A* – right). There was a concomitant decrease in the average distance of the point of gaze from the goal location in a step-like manner with each subgoal achieved (*Figure 5B* – right). In contrast to backward sweeps, which were made predominantly to the most proximal subgoal prior to navigating (*Figure 5C* – left), forward sweeps that occurred during movement were not regularly directed

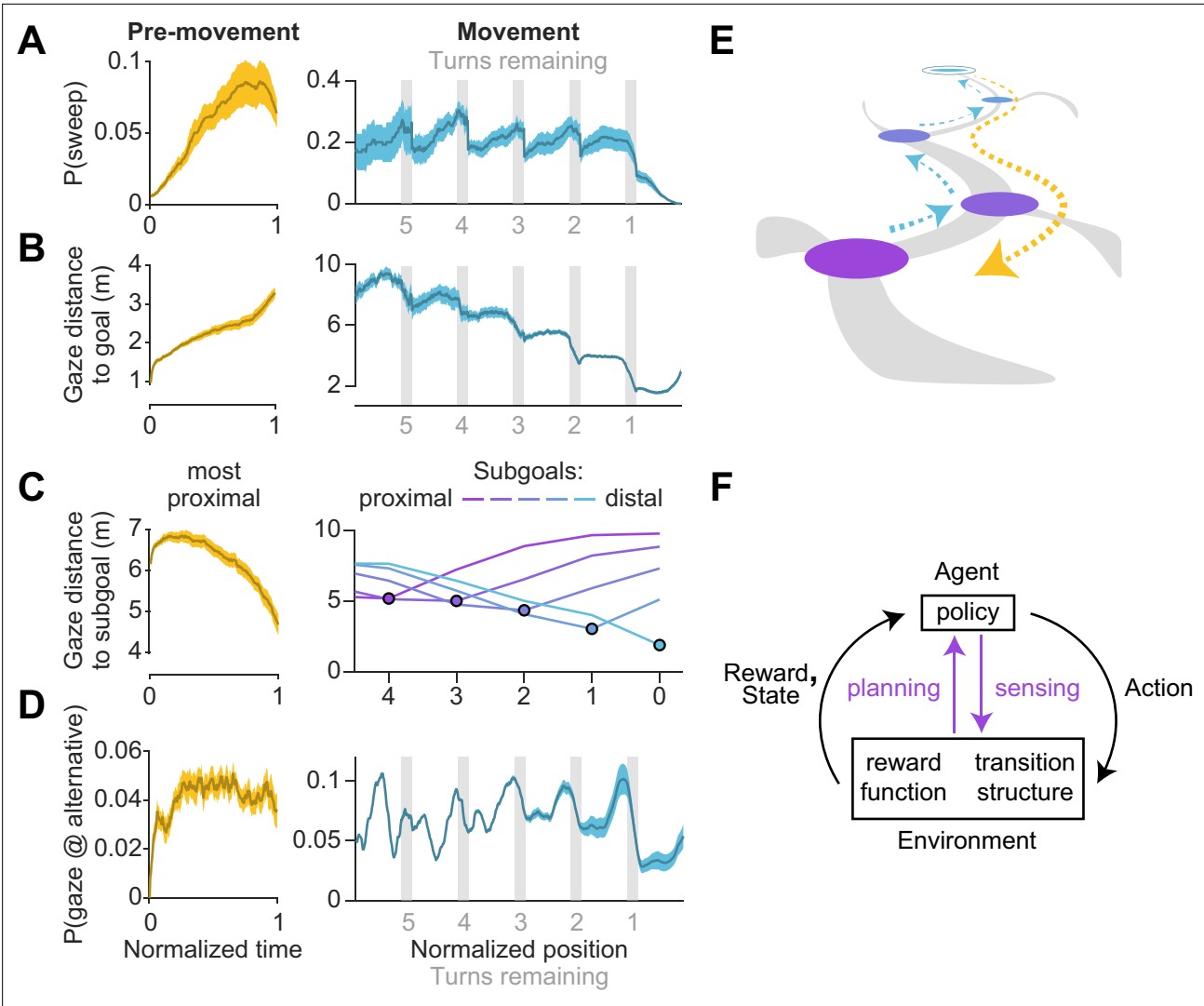

**Figure 5.** Timing of sweeps reveals task decomposition. Trials across all arenas and all participants were aligned and scaled for the purpose of trial-averaging. This process was carried out separately for the pre-movement and movement epochs. (**A**) Left: Prior to movement, the probability of (backward) sweeps increased with time. Right: During movement, the probability of (forward) sweeps transiently increased at the precise moments when participants reached each subgoal. Participant position is defined in relation to the location of subgoals. Subgoals are designated as numbers starting from the goal (subgoal 0) and counting backwards along the trajectory (subgoals 1, 2, 3 etc.) such that greater values correspond to more proximal subgoals. (**B**) Left: Gaze traveled away from the goal location prior to movement. Right: The average distance of gaze from the goal decreased in steps, with steps occurring at each subgoal. (**C**) Distance of gaze from individual subgoals (most proximal in yellow, most distal in cyan). Left: Gaze traveled towards the most proximal subgoal prior to movement, consistent with the increased probability of backward sweeps during this epoch. Right: The average distance of gaze to each individual subgoal (colored lines) was minimized precisely when participants approached that subgoal. (**D**) Left: The probability of gazing at alternative trajectories is relatively constant throughout the pre-movement epoch. Right: Participants gaze at alternative trajectories more frequently when approaching turns. (**E**) A graphical summary of the spatiotemporal dynamics of eye movements in this task. Subgoals are depicted in the same color scheme used in **C**. (**F**) Diagram of a standard Markov Decision Process, augmented with an additional pathway for agent-environment interaction through eye movements (colored arrows). Dashed arrows denote sweeps, and possible paths throughout the arena are depicted in gray. Darker bounds in **A**–**C** denote ±1 SEM.

toward one particular location. Instead, in a strikingly stereotyped manner, participants appeared to lock their gaze upon the *upcoming* subgoal when rounding each bend in the trajectory (*Figure 5C* – right). This suggests that participants likely represented their plan by decomposing it into a series of subgoals, focusing on one subgoal at a time until they reached the final goal location. In contrast to sweeping eye movements, the likelihood of gazing alternative trajectories peaked much earlier during pre-movement (*Figure 5D* – left). Likewise, during movement, participants tend to look briefly

at alternative trajectories shortly before approaching a subgoal (*Figure 5D* – right) which might constitute a form of vicarious trial and error behavior at choice points (*Redish, 2016*).

To summarize, we found participants made sequential eye movements sweeping forward and/or backward along the intended trajectory, and the likelihood of sweeping increased with environmental complexity. During the pre-movement phase, participants gathered visual information about the arena, evaluated alternative trajectories, and typically traced the chosentrajectory backwards from the goal to the first subgoal (*Figure 5E* – orange). While moving through the arena, they tended to lock their gaze upon the upcoming subgoal until shortly before a turn, at which point they exhibit a higher tendency of gazing upon alternative trajectories. During turns, participants often sweep their gaze forward to the next subgoal (*Figure 5E* – blue). Via eye movements, navigators could construct well-informed plans to make sequential actions that would most efficiently lead to rewards (*Figure 5F*).

## Discussion

In this study, we highlight the crucial role of eye movements for flexible navigation. We found that humans took trajectories nearly optimal in length through unfamiliar arenas, and spent more time planning prior to navigation in more complex environments. The spatial distribution of gaze was largely concentrated at the hidden goal location in the simplest environment, but participants increasingly interrogated the task-relevant structure of the environment as the arena complexity increased. In the temporal domain, participants often rapidly traced their future trajectory to and from the goal with their eyes (sweeping), and generally concentrated their gaze upon one subgoal (turn) at a time until they reached their destination. In summary, we found evidence that the neural circuitry governing the oculomotor system optimally schedules and allocates resources to tackle the diverse cognitive demands of navigation, producing efficient eye movements through space and time.

Eye movements provide a natural means for researchers to understand information seeking strategies, in both experimental and real-world settings (*Gottlieb et al., 2013*; *Gottlieb et al., 2014*). Past studies using simple decision making tasks probed whether active sensing, specifically via eye movements, reduces uncertainty about the *state* of the environment (such as whether a change in an image has occurred) (*Yang et al., 2016*; *Renninger et al., 2007*; *Ahmad and Yu, 2013*). But in sequential decision making tasks such as navigation, there is added uncertainty about the task-contingent causal structure (*model*) of the environment (*Kaplan and Friston, 2018*; *Mattar and Lengyel, 2022*). Common paradigms for goal-oriented navigation occlude large portions of the environment from view, usually in the interest of distinguishing model-based strategies from conditioned responses (*Smittenaar et al., 2013*; *Simon and Daw, 2011*; *de Cothi et al., 2020*) or allow very restricted fields of view where eye movements have limited potential for sampling information (*Javadi et al., 2017*; *Ghamari and Golshany, 2021*). Occlusions eliminate the possibility of gathering information about the structure of the environment using active sensing. By removing such constraints, we allowed participants to acquire a model of the environment without physically navigating through it, which yielded new insights about how humans perform visually-informed planning.

In particular, we found that the gaze is distributed between the two components of the model required to plan a path – the transition function, which describes the relationship between states, and the reward function, which describes the relationship between the states and the reward – with the distribution skewed in favor of the former in more complex environments. When alternative paths were available, gaze tended to be directed towards them at the expense of looking at the hidden reward location. These findings suggest a context-dependent mechanism which dictates the dynamic arbitration between competing controllers of the oculomotor system that seek information about complementary aspects of the task. Neurally, this could be implemented by circuits that exert executive control over voluntary eye movements. Candidate substrates include the dorsolateral prefrontal cortex, which is known to be important for contextual information processing and memory-guided saccades (*Pierrot-Deseilligny et al., 1995*; *Johnston and Everling, 2006*; *Pierrot-Deseilligny et al., 2005*), and the anterior cingulate cortex, which is known to be involved in evaluating alternative strategies (*Tervo et al., 2021*; *Gaymard et al., 1998*). To better understand the precise neural mechanisms underlying the spatial gaze patterns we observed, it would be instructive to examine the direction of information flow between the oculomotor circuitry and brain regions with strong spatial and value representations during this task in animal models. Future research may also investigate multi-regional interactions in humans by building on recent advances in data analysis that allow for eye movements

to be studied in fMRI scanners (*Frey et al., 2021*). Our analysis of spatial gaze patterns is grounded in the RL framework, which provides an objective way to measure the utility of sampling information from different locations. However, this measure was agnostic to the temporal ordering of those samples. Given that previous work demonstrated evidence for the planning of multiple saccades during simple tasks like visual search (*Hoppe and Rothkopf, 2019*), incorporating chronology into a normative theory of eye movements in sequential decision-making tasks presents an excellent opportunity for future studies.

Meanwhile, we found that the temporal pattern of eye movements revealed a fine-grained view of how planning computations unfold in time. In particular, participants made sequential eye movements sweeping forward and/or backward along the future trajectory, evocative of forward and reverse replay by place cells in the hippocampus (*Diba and Buzsáki, 2007*; *Pfeiffer and Foster, 2013*). Shortly after fixating on the goal, participants' gaze often swept backwards along their future trajectory, mimicking reverse replay. Because these sweeps predominantly occurred before movement, they may reflect depth-first tree search, a model-based algorithm for path discovery (*Zhou and Hansen, 2008*). Then, during movement, participants were more likely to make forward sweeps when momentarily slowing down at turning points, analogous to finding that neural replay mainly occurs during periods of relative immobility (*Sosa and Giocomo, 2021*). Several recent studies have also supported that replay serves to consolidate memory and generalize information about rewards (*Gillespie et al., 2021*; *Liu et al., 2021*; *Eldar et al., 2020*). In light of the similarities between sweeps and sequential hippocampal activations, we predict that direct or indirect hippocampal projections to higher oculomotor controllers (e.g. the supplemental eye fields through the orbitofrontal cortex) may allow eye movements to embody the underlying activations of state representations (*Larson and Loschky, 2009*; *Wilming et al., 2017*; *Hannula and Ranganath, 2009*). This would allow replays to influence the active gathering of information. Alternatively, active sensing could be a result of rapid peripheral vision processing which drives saccade generation, such that the eye movements reflect the outcome of sensory processing rather than prior experience. Consistent with this idea, past studies have demonstrated that humans can smoothly trace paths through entirely novel 2D mazes (*Crowe et al., 2000*; *Crowe et al., 2004*). Interestingly, neural modulation does occur in this direction — the contents of gaze have been found to influence activity in the hippocampus and entorhinal cortex (*Turk-Browne, 2019*; *Liu et al., 2017*; *Monaco et al., 2014*; *Jun et al., 2016*; *Fotowat et al., 2019*; *Ringo et al., 1994*). Therefore, it is conceivable that sequential neural activity could emerge from consolidating temporally extended eye movements such as sweeps. We hope that in future, simultaneous recordings from brain areas involved in visual processing, eye movement control, and the hippocampal formation would uncover the mechanisms underlying trajectory sweeping eye movements and their relationship to perception and memory.

Value-based decisions are known to involve lengthy deliberation between similar alternatives (*Bakkour et al., 2019*; *Tajima et al., 2016*). Participants exhibited a greater tendency to deliberate between viable alternative trajectories at the expense of looking at the reward location. Likelihood of deliberation was especially high when approaching a turn, suggesting that some aspects of path planning could also be performed on the fly. More structured arena designs with carefully incorporated trajectory options could help shed light on how participants discover a near-optimal path among alternatives. However, we emphasize that deliberative processing accounted for less than one-fifth of the spatial variability in eye movements, such that planning largely involved searching for a viable trajectory.

Although we have analyzed strategies of active sensing and planning separately, these computations must occur simultaneously and influence each other. This is formalized by the framework of active inference that unifies planning and information seeking by integrating the RL framework, which describes exploiting rewards for their extrinsic value, and the information theoretic framework, which describes exploring new information for its epistemic value (*Kaplan and Friston, 2018*). Using this framework to simulate eye movements in a spatial navigation task, *Kaplan and Friston, 2018* found that gaze is dominated by epistemic (curiosity) rather than pragmatic (reward) considerations in the first few trials, a prediction that is not supported by our results. However, it is possible that participants were able to rapidly resolve uncertainty about the arena structure in our experiments. Future studies must identify the constraints under which active inference models can provide quantitatively good fits to our data. In another highly relevant theoretical work, Mattar and Daw proposed that

path planning and structure learning are variants of the same operation, namely the spatiotemporal propagation of memory (*Mattar and Daw, 2018*). The authors show that prioritization of reactivating memories about reward encounters and imminent choices depends upon its utility for future task performance. Through this formulation, the authors provided a normative explanation for the idiosyncrasies of forward and backward replay, the overrepresentation of reward locations and turning points in replayed trajectories, and many other experimental findings in the hippocampus literature. Given the parallels between eye movements and patterns of hippocampal activity, it is conceivable that gaze patterns can be parsimoniously explained as an outcome of such a prioritization scheme. But interpreting eye movements observed in our task in the context of the prioritization theory requires a few assumptions. First, we must assume that traversing a state space using vision yields information that has the same effect on the computation of utility as does information acquired through physical navigation. Second, peripheral vision allows participants to form a good model of the arena such that there is little need for active sensing. In other words, eye movements merely reflect memory access and have no computational role. Finally, long-term statistics of sweeps gradually evolve with exposure, similar to hippocampal replays. These assumptions can be tested in future studies by titrating the precise amount of visual information available to the participants, and by titrating their experience and characterizing gaze over longer exposures. We suspect that a pure prioritization-based account might be sufficient to explain eye movements in relatively uncluttered environments, whereas navigation in complex environments would engage mechanisms involving active inference. Developing an integrative model that features both prioritized memory-access as well as active sensing to refine the contents of memory, would facilitate further understanding of computations underlying sequential decision-making in the presence of uncertainty.

The tendency of humans to break larger problems into smaller, more tractable subtasks has been previously established in domains outside of navigation (*Killian et al., 2012*; *Meister and Buffalo, 2018*; *Killian and Buffalo, 2018*; *Eckstein and Collins, 2020*; *Tomov et al., 2020*). However, theoretical insights on clustered representations of space have not been empirically validated in the context of navigation (*Balaguer et al., 2016*; *Rasmussen et al., 2017*), primarily due to the difficulty in distinguishing between flat and hierarchical representations from behavior alone. Our observation that participants often gazed upon the upcoming turn during movement supports that participants viewed turns as subgoals of an overall plan. Future work could focus on designing more structured arenas to experimentally separate the effects of path length, number of subgoals, and environmental complexity on participants' eye movement patterns.

We hope that the study of visually-informed planning during navigation will eventually generalize to understanding how humans accomplish a variety of sequential decision-making tasks. A major goal in the study of neuroscience is to elucidate the principles of biological computations which allow humans to effortlessly exceed the capabilities of machines. Such computations allow animals to learn environmental contingencies and flexibly achieve goals in the face of uncertainty. However, one of the main barriers to the rigorous study of active, goal-oriented behaviors is the complexity in estimating the participant's prior knowledge, intentions, and internal deliberations which lead to the actions that they take. Luckily, eye movements reveal a wealth of information about ongoing cognitive processes during tasks as complex and naturalistic as spatial navigation.

## Methods
### Experimental Model and Participant Details

Thirteen human participants (all >18 years old, ten males) participated in the experiments. All but two participants (S6 and S9) were unaware of the purpose of the study. Four of the participants, including S6 and S9, were exposed to the study earlier than the rest of the participants, and part of the official dataset for two of these participants (S4 and S8) was collected 2 months prior to the rest of data collection as a safety precaution during the COVID-19 pandemic. Eight additional human participant recruits (all >18 years old, four males) were disqualified due to experiencing motion sickness while in the VR environment and not completing a majority of trials. All experimental procedures were approved by the Institutional Review Board at New York University and all participants signed an informed consent form (IRB-FY2019-2599).

## Stimulus

Participants were seated on a swivel chair with 360° of freedom in physical rotation and navigated in a full-immersion hexagonal virtual arena with several obstacles. The stimulus was rendered at a frame rate of 90 Hz using the Unity game engine v2019.3.0a7 (programmed in C#) and was viewed through an HTC VIVE Pro virtual reality headset. The subjective vantage point (height of the point between the participants' eyes with respect to the ground plane) was 1.72 m. The participant had a field of view of 110.1° of visual angle. Forward and backward translation was enabled via a continuous control CTI Electronics M20U9T-N82 joystick with a maximum speed recorded at 4.75 m/s. Participants executed angular rotations inside the arena by turning their head, while the joystick input enabled translation in the direction in which the participant's head was facing. Obstacles and arena boundaries appeared as gray, rectangular slabs of concrete. The ground plane was grassy, and the area outside of the arena consisted of a mountainous background. Peaks were visible above the outer boundary of the arena to provide crude orientation landmarks. Clear blue skies with a single light source appeared overhead.

## State space geometry

The arena was a rectangular hexagon enclosing an area of approximately 260 m² of navigable space. For ease of simulation and data analyses, the arena was imparted with a hidden triangular tessellation (*deltille*) composed of 6 $n^2$ equilateral triangles, where n determines the state space granularity. We chose n=5, resulting in triangles with a side length of 2 meters, each of which constituted a state in the discrete state space (*Figure 1—figure supplement 1A*). The arena contained several obstacles in the form of unjumpable obstacles (0.4 m high) located along the edges between certain triangles (states). Obstacle locations were predetermined offline using MATLAB by either randomly selecting a chosen number of edges of the tessellation or by using a graphical user interface (GUI) to manually select edges of the tessellation; these locations were loaded into Unity. Outer boundary walls of height 2.5 m enclosed the arena. We chose five arenas spanning a large range in *average* state closeness centrality $\langle C(s) \rangle$ (*Equation 2*), where $C(s)$ is defined as the inverse average path length $d$ from state $s$ to every other state $s'$ ($N$ states in total). On average, arenas with lower centrality will impose greater path lengths between two given states, making them more complex to navigate. We defined a measure of arena complexity by adding an offset to mean centrality and then scaling it, such that the simplest arena had a complexity value of zero ($-100 \times (\bar{C} - \max[\bar{C}])$) where $\bar{C}$ denotes mean centrality across states and max is taken over arenas. Such a transformation would preserve the correlation and p-values between dependent variables and arena centrality, while the new metric would allow for the graphic representation of arenas in an intuitive order (*Figure 1B*). A complexity value of zero corresponds to the simplest arena that we designed. The order of arenas presented to each participant was randomly permuted but not entirely counterbalanced due to the large number of permutations (*Appendix 2—table 1*).

$$C(s) = \frac{N-1}{\sum_{s'} d(s',s)} \qquad (2)$$

## Eye tracking

At the beginning of each block of trials, participants calibrated the VIVE Pro eye tracker using inbuilt Tobii software which prompted participants to foveate several points tiling a 2D plane in the VR environment. Both eyes were tracked, and the participant's point of foveation (*x-y* coordinates), object of foveation (ground, obstacles, boundaries, etc.), eye openness, and other variables of interest were recorded on each frame using the inbuilt software. *Sipatchin et al., 2020* reported that during free head movements, point-of-gaze measurements using the VIVE Pro eye tracker has a spread of 1.15° ± 0.69° (SE) (*Sipatchin et al., 2020*). This means that when the participant fixates a point on the ground five meters away, the 95% confidence interval (CI) for the measurement error in the reported gaze location would be 0–23 cm (roughly one-tenth of the length of one transition or obstacle) and 0–67 cm (one-third of a transition length) for points fifteen meters away. While machine precision was not factored into the analyses, the fraction of eye positions that may have been misclassified due to hardware and software limitations is likely very small. Furthermore, Sipatchin et al. reported that the system latency was 58.1 ms. While there is reason to suspect that the participant's position was recorded with a similar latency of around 5 frames, even if the gaze data lagged the position data, the

participant would only have moved 28.5 cm if they were translating at the maximum possible velocity over this interval.

## Behavioral task

At the beginning of each trial, a target in the form of a realistic banana from the Unity Asset store appeared hovering 0.4 m over a state randomly drawn from a uniform distribution over all possible states. The joystick input was disabled until the participant foveated the target, but the participant was free to scan the environment by rotating in the swivel chair during the visual search period. About 200 ms after target foveation, the banana disappeared and participants were tasked with navigating to the remembered target location without time constraints. Participants were not given instructions on what strategy to use to complete the task. After reaching the target, participants pressed a button on the joystick to indicate that they have completed the trial. Alternatively, they could press another button to indicate that they wished to skip the trial. Feedback was displayed immediately after pressing either button (see section below). Skipping trials was discouraged except when participants did not remember seeing the target before it disappeared, and these trials were recorded and excluded from the analyses (< 1%).

## Reward

If participants stopped within the triangular state which contained the target, they were rewarded with two points. If they stopped in a state sharing a border with the target state, they were rewarded with one point. After the participant's button press, the number of points earned on the current trial was displayed for one second at the center of the screen. The message displayed was 'You earned p points!'; the font color was blue if p=1 or p=2, and red if p=0. On skipped trials, the screen displayed 'You passed the trial' in red. In each experimental session, after familiarizing themselves with the movement controls by completing ten trials in a simplistic six-compartment arena (granularity n=1), participants completed one block of 50 trials in each of five arenas (*Figure 1B*). At the end of each block, a blue message stating 'You have completed all trials!' prompted them to prepare for the next block. Session durations were determined by the participant's speed and the length of the breaks that they needed from the virtual environment, ranging from 1.5–2 hr, sometimes spread across more than 1 day. Participants were paid $0.02/point for a maximum of 5 arenas × 50 trials/arena × 2 points/trial × $0.02/point = $10, in addition to a base pay of $10 /hr for their time (the average payment was $27.55).

## RL formulation

Navigation can be formulated as a Markov decision process (MDP) described by the tuple $< S, A, P, R, \gamma >$ whose elements denote, respectively, a finite state space $S$, a finite action space $A$, a state transition distribution $P$, a reward function $R$, and a temporal discount factor γ that captures the relative preference of distal over proximal rewards (*Bermudez-Contreras et al., 2020*). Given that an agent is in state $s \in S$, the agent may execute an action $a \in A$ in order to bring about a change in state $s \rightarrow s'$ with probability $P(s'|s, a)$ and harvest a reward $R(s, a)$. To relate this formalism to the structure of the arena, it is instructive to consider the *possibility* of traversal from state $s$ to any state $s'$ in a single time step, as described by the adjacency matrix $T$ if there exists an available action which would bring about the change in state $s \rightarrow s'$ with a non-zero probability, and $T(s, s') = 0$ otherwise. By definition, $T(s, s') = 0$ if there is an obstacle between $s$ and $s'$. A state not bordered by any obstacle would have three non-zero entries in the corresponding row of $T$. Thus, the arena structure is fully encapsulated in the adjacency matrix.

In the case that an agent is tasked with navigating to a goal location $s_G$ where the agent would receive a reward, the reward function $R(s, a) > 0$ if and only if the action $a$ allows for the transition $s \rightarrow s_G$ in one time step, and $R(s, a) = 0$ otherwise. Given this formulation, we may compute the optimal policy $\pi^*(a|s)$, which describes the actions that an agent should take from each state in order to reach the target state in the fewest possible number of time steps. The optimal policy may be derived by computing optimal state values $V^*(s)$, defined as the expected future rewards to be earned when an agent begins in state $s$ and acts in accordance with the policy $\pi^*$. The optimal value function can be computed by solving the Bellman Equation (*Equation 3*) via dynamic programming (specifically value iteration) — an efficient algorithm for path-finding — which iteratively unrolls the recursion in

this equation (*Bellman, 1954*). The optimal policy is given by the argument $a$ that maximizes the right-hand side of *Equation 3*. Intuitively, following the optimal policy requires that agents take actions to ascend the value function where the value gradient is most steep (*Figure 1D*).

$$V^*(s) = \max_a [R(s,a) + \gamma \sum_{s'} P(s'|s,a)V^*(s')] \qquad (3)$$

For the purposes of computing the optimal trajectory, we considered twelve possible degrees of freedom in the action space, such that one-step transitions could result in relocating to a state that is 0°, 30°, 60°, …, 300°, or 330° with respect to the previous state. However, the center-to-center distances between states for a given transition depends on the angle of transition. Specifically, as shown in *Figure 1—figure supplement 1C*, if a step in the 0° direction requires translating 1 m, then a step in the 60°, 120°, 180°, 240°, and 300° directions would also require translating 1 m, but a step in the 30°, 150°, and 270° directions would require translating $2\sqrt{3}/3$ m, and a step in the 90°, 210°, and 330° directions would require translating $\sqrt{3}/3$ m. Therefore, in *Equation 3*, $R(s,a) = -1, -2\sqrt{3}/3$, or $-\sqrt{3}/3$, depending on the step size required in taking an action $a$. The value of the goal state $s_G$ was set to zero on each iteration. Value functions were computed for each goal location, and the relative value of states describes the relative minimum number of time steps required to reach $s_G$ from each state. The lower the value of a state, the greater the geodesic separation between the state and the goal state. We set $\gamma = 1$ during all simulations and performed 100 iterations before calculating optimal trajectory lengths from an initial state $s_i$ to the target state $s_G$, as this number of iterations allowed for the algorithm to converge.

## Relevance computation

To compute the relevance $\Omega_k(s_0, s_G)$ of the $k^{th}$ transition to the task of navigating from a specific initial state $s_0$ to a specific goal $s_G$, we calculated the absolute change induced in the optimal value of the initial state after toggling the navigability of that transition by changing the corresponding element in the adjacency matrix from 1 to 0 or from 0 to 1 (*Equation 4*). For the simulations described in Appendix 1, we also tested a non-myopic, path-dependent metric $\Omega_k(s_0, s_G; \pi^*)$ defined as the sum of squared differences induced in the values of all states along the optimal path (*Equation 5*). Furthermore, we tested the robustness of the measure to the precise algorithm used to compute state values by computing value functions using the successor representation (SR) algorithm, which caches future state occupancy probabilities learned with a specific policy (*Stachenfeld et al., 2017*). (While SR is more efficient than value iteration, it is less flexible.) As we used a random walk policy, we computed the matrix of probabilities $M$ analytically by temporally abstracting a one-step transition matrix $T$. The cached probabilities can then be combined with a one-hot reward vector $R(s) = \mathbb{1}(s = s_G)$ to yield state values $V = MR$. We set the temporal discount factor $\gamma = 1$ and integrated over 100 time steps.

$$\Omega_k(s_0, s_G) = [V(s_0|T_k = 1) - V(s_0|T_k = 0)]^2 \qquad (4)$$

$$\Omega_k(s_0, s_G; \pi^*) = \sum_{s=s_0}^{s_G} [V(s|T_k = 1) - V(s|T_k = 0)]^2 \qquad (5)$$

## Relation to bottlenecks

In order to assess whether the relevance metric is predictive of the degree to which transitions are bottlenecks in the environment, we correlated normalized relevance values (averaged across all target locations and normalized via dividing by the maximum relevance value across all transitions for each target location) with the average betweenness centrality $G$ of the two states on either side of a transition (*Equation 6*). Betweenness centrality essentially calculates the degree to which a state controls the traffic flowing through the arena. $\sigma_{ij}$ represents the number of shortest paths between states $i$ and $j$, and $\sigma_{ij}(s)$ represents the number of such paths which pass through state $s$. For this analysis, transitions within 1 m from the goal state were excluded due to their chance of having spuriously high relevance values.

$$G(s) = \sum_{i \neq s \neq j} \frac{\sigma_{ij}(s)}{\sigma_{ij}} \qquad (6)$$

## Simulations

Behavior of three artificial agents with qualitatively different planning capacities was simulated. All agents were initialized with a noisy model of the environment. Representational noise was simulated by toggling 50% of randomly selected unavailable transitions from $T(s, s') = 0$ to 1, and the equivalent number of randomly selected available transitions from $T(s, s') = 1$ to . This is analogous to the agents misplacing obstacles in their memories, or equivalently, a subjective-objective model mismatch induced by volatility in the environment. The blind agent was unable to correct its model during a planning period. On each trial, eight transitions (out of 210 available) were drawn for each sighted agent; the agent's model was compared with the true arena structure at these transitions and, if applicable, corrected prior to navigation. Visual samples were drawn uniformly from all possible transitions (without replacement) for the random exploration agent. For the goalward looking agent, the probability of drawing a transition was determined by a circular normal (von Mises) distribution with $\mu = \theta_G$ (where $\theta_G$ is the angle of the goal w.r.t. the agent's heading), $\sigma = 1$, and concentration parameter $\kappa = 5$. In contrast, the directed sampling agent gathered information specifically about the eight transitions that were calculated to be most relevant for that trial. After the model updates, if any, the agents' subjective value functions were recomputed, and agents took actions according to the resulting policies. When an agent encountered a situation in which no action was subjectively available, they attempted a random action. In the case that a new action is discovered, the agents temporarily updated $T(s, s')$ from 0 to 1 for that action. Conversely, in the case that an agent attempted to take an action but discovered that it was not actually feasible, they temporarily updated their subjective models to account for the transition block which they had just learned about. In both cases, value functions were recomputed using the updated model. Simulations were conducted with 25 arenas of granularity n=3 (state space size = 54 for computational tractability) and 100 trials per arena. Furthermore, we tested the agents' performance using a range of gaze samples evenly spaced between 2 and 14 foveations.

## Data processing

In order to identify moving and non-moving epochs within each trial, movement onset and offset times were detected by applying a moving average filter of window size 5 frames on the absolute value of the joystick input function. When the smoothed joystick input exceeded the threshold of 0.2 m/s (approximately 10% of the maximum velocity), the participant was deemed to be moving, and when the input fell below this threshold for the last time on each trial, the participant was deemed to have stopped moving. Participants' relative planning time was defined as the ratio of pre-movement time to the total trial duration, minus the search period (which was roughly constant across arenas). Prior to any eye movement analyses, blinks were filtered from the eye movements by detecting when the fraction of the pupil visible dipped below 0.8. The spread in the $(x, y)$ gaze positions within trials was calculated as the expectation of variance, $\mathbb{E}_n[\sqrt{\text{Var}_t[x] + \text{Var}_t[y]}\,]$, where $\text{Var}_t[\cdot]$ denotes the variance across time $t$ within a trial and $\mathbb{E}_n[\cdot]$ denotes expectation across trials denoted by $n$. The spread across trials was calculated as the variance of the expectation, $\sqrt{\text{Var}_n[\mathbb{E}_t[x]] + \text{Var}_n[\mathbb{E}_t[y]]}$, where $\mathbb{E}_t[\cdot]$ denotes expectation across time and $\text{Var}_n[\cdot]$ denotes the variance across trials.

For *Figure 1f, g*, *Figure 1—figure supplement 2A, B, E* the first trial of each run was removed from the analyses due to an occasional rapid teleportation of the participant to a random starting location associated with the software starting up. While there were a few instances where more than one run occurred per block due to participants adjusting the headset, at least 51 trials were actually collected during each block such that most blocks consisted of 50 trials when the first trial of each run was omitted. For analyses such as epoch duration, gaze distribution, relevance, sweep detection, and subgoal detection, the first trial was not discarded since the teleportation only affected the recorded path length, but as the teleportation was virtually instantaneous, the new starting locations on such trials could be used for analyses which do not depend upon the path length variable.

## Linear mixed effects models

To separately examine how various aspects of the navigation task contribute to the behavioral and eye movement patterns that we observed, we fit linear mixed effects models of the form *Equation 7*, where each datapoint either corresponds to one trial (for trial-level analyses) or one participant in one arena (for analyses like % trials). In the former case, the number of predictor variables was $J = 4$, and

a fixed effect slope $\beta_j$ and participant-specific random effect slope $\beta_{ij}$, quantified the effect of each variable in a linear combination with a fixed intercept $\beta_0$ and a participant-specific random intercept $\beta_{i0}$ to produce an estimate of the dependent variable $y_i$ on that trial for participant . The three variables were the number of turns (described in the 'Subgoal analysis' section), the length of the optimal trajectory (described in the 'RL formulation' section), and the unsigned angle between the direction of participants' initial heading vs the optimal direction of target approach (relative bearing). The ranges of the predictors were – number of turns: 0–17, length of optimal trajectory: 0–45 m, relative bearing: 0–90 degrees. *Equation 8* describes the specific form used to examine trial-specific effects on the output. For the case where analyses required pooling trials for each participant in each arena, $J = 1$ and the single set of fixed/random slopes correspond to the arena complexity variable. All variables were z-scored for each participant prior to model fittings, so the intercepts were close to zero for most cases and not shown in the bar plots.

$$y_i = \beta_0 + \beta_{i0} + \sum_{j=1}^{J}(\beta_j x_j + \beta_{ij} x_j) \tag{7}$$

$$\text{Output} \sim (1|\text{Participant}) + \text{NTurns} + (\text{NTurns} - 1|\text{Participant}) + \text{PathLength} + (\text{PathLength} - 1|\text{Participant})$$

$$+(\text{Bearing} - 1|\text{Participant})$$

$$\tag{8}$$

## Relevance estimation

Prior to estimating the task-relevance of the participants' gaze positions at each time point, the closest transition $k$ to the participant's point of gaze was identified and the effect of toggling the transition on the value function was computed as $\Omega_k(s(t), s_G)$. In order to construct a null distribution of relevance values, we paired the eye movements on each trial with the goal location for a random trial, given the participant's position in the current trial. This shuffled average is not task-specific, and therefore may be compared with the true $\Omega$ values to probe whether the spatial distribution of gaze positions was sensitive to the goal location on each trial. Similarly, the shuffled fraction of time looking at the goal was computed with a goal state randomly chosen from all states.

## Sweep classification

Forward and backward eye movements (sweeps) along the intended trajectory were classified by first calculating the point $(x, y)$ on the trajectory closest to the location of gaze in each frame. For each trial, the fraction of the total trajectory length corresponding to each point was stored as a variable $f$, and periods when $f(t)$ consecutively ascended or descended were identified. For each period, we determined $m$, an integer whose magnitude denoted the sequence length and whose sign denoted the sequence direction (+/− for ascending/descending sequences). We then constructed a null distribution $P(m)$ describing the chance-level frequency of $m$ by selecting 20 random trials and recomputing $f$ based on the participant's trajectories on those trials. Sequential eye movements of length $m$ where the CDF of $P(m)$ was less than $\alpha/2$ or greater than $1 - \alpha/2$ were classified as backward and forward sweeps, respectively. The significance threshold α was chosen to be 0.02. Compensating for noise in the gaze position, we applied a median filter of length 20 frames to both the true and shuffled $f$ functions. During post-processing, sweeps in the same direction that were separated by less than 25 frames were merged, and sweeps for which the gaze fell outside of 2 meters from the intended trajectory on > 30% of the frames pertaining to the sweep were eliminated. Sweeps were required to be at least 25 frames in length. To remove periods of fixation, the minimum variance in $f(t)$ values for all time points corresponding to the sweep was required to be 0.001. Finally, sweeps which did not cover at least 20% of the total trajectory length were removed from the analyses. This algorithm allowed for the automated detection of sequential eye movements pertaining to the prospective evaluation of trajectories which participants subsequently took.

## Alternative trajectories

To find the number of trajectory options for each trial, we identified all paths between the initial and goal states that were comparable within a factor of 1.25 to the optimal trajectory length and shared no more than 50% of the states with each other. The factor of 1.25 ensured that the trajectories were within 1 SD of the trajectory chosen by the participants. The gaze was classified to be exploring an alternative trajectory only when it was disjoint from the trajectory that the participant executed.

## Saccade detection

Saccade times were classified to be the times at which eye movement speeds $v$ crossed a threshold of 50 °/s from below, where speeds were computed using *Equation 9*, where $x$, $y$, and $z$ correspond to the coordinates of the point of gaze (averaged across both eyes), and α and β respectively correspond to the lateral and vertical displacement of the pupil in degrees.

$$\alpha(t) = \tan^{-1}\left(\frac{x(t)}{\sqrt{y^2(t)+z^2(t)}}\right), \ \beta(t) = \tan^{-1}\left(\frac{z(t)}{\sqrt{y^2(t)+x^2(t)}}\right), \ v(t) = \sqrt{\left(\frac{d}{dt}\alpha(t)\right)^2 + \left(\frac{d}{dt}\beta(t)\right)^2} \tag{9}$$

## Subgoal analysis

Turns in the participants' trajectories (defined as subgoals) were isolated by applying a threshold of 60 °/s on their angular velocity (smoothed with a median filter; window size = 8 frames). The first and last frames for periods of elevated angular velocity were recorded. For the purposes of the analysis in *Figure 5*, all trials (for all participants in all arenas) were broken down into periods of turns vs periods of navigating straight segments. Starting from the stopping location, these periods were independently interpolated to fit an arbitrarily defined common timeline of 25 time points per turn and 100 time points per straight segment. For example, if a trial had two turns, then eye movement variables from the last turn to the stopping location were interpolated to the points –100 to –1. The last turn was represented by –125 to –101, and the second last turn was represented by –250 to –226. The segment between the two turns was represented by –225 to –126. Finally, the segment between the participant's starting position and the first turn was represented by –350 to –251. Note that as a consequence, the number of trials for which there were (for example) more than four turns in the trajectory was substantially fewer than the number of trials for which there were one or no turns, such that the quantity of raw data contributing to each normalized position value in *Figure 5* increases from left to right.

## Data and code availability

The dataset is available at https://gin.g-node.org/neuro-sci/gaze-navigation, and the MATLAB code used to produce the analysis figures has been published at https://github.com/neuro-sci/gaze-navigation, (copy archived at swh:1:rev:91870d7384c539b656f5dcab69bc24b83eece161; *Zhu, 2022*).

## Acknowledgements

We thank the members of the Angelaki Lab and Professor Wei Ji Ma for insightful discussions. This work was supported by the NIH (1U19-NS118246 – BRAIN Initiative, 1R01-EY022538), the NSF NeuroNex Award (DBI-1707398) and the Gatsby Charitable Foundation.

## Additional information

### Funding

| Funder | Grant reference number | Author |
| --- | --- | --- |
| National Institutes of Health | U19-NS118246 | Seren Zhu<br>Nastaran Arfaei<br>Dora E Angelaki |
| National Institutes of Health | R01-EY022538 | Seren Zhu<br>Nastaran Arfaei<br>Dora E Angelaki |
| National Science Foundation | DBI-1707398 | Kaushik Janakiraman<br>Lakshminarasimhan |
| Gatsby Charitable Foundation | | Kaushik Janakiraman<br>Lakshminarasimhan |

The funders had no role in study design, data collection and interpretation, or the decision to submit the work for publication.

## Author contributions
Seren Zhu, Conceptualization, Data curation, Formal analysis, Investigation, Methodology, Software, Visualization, Writing – original draft, Writing – review and editing; Kaushik J Lakshminarasimhan, Conceptualization, Formal analysis, Methodology, Supervision, Visualization, Writing – original draft, Writing – review and editing; Nastaran Arfaei, Conceptualization, Methodology, Software; Dora E Angelaki, Conceptualization, Funding acquisition, Methodology, Project administration, Resources, Supervision, Writing – review and editing

## Author ORCIDs
Seren Zhu http://orcid.org/0000-0003-0555-9690
Kaushik J Lakshminarasimhan http://orcid.org/0000-0003-3932-2616
Dora E Angelaki http://orcid.org/0000-0002-9650-8962

## Ethics
Human subjects: All experimental procedures were approved by the Institutional Review Board at New York University and all participants signed an informed consent form (IRB-FY2019-2599).

## Decision letter and Author response
Decision letter https://doi.org/10.7554/eLife.73097.sa1
Author response https://doi.org/10.7554/eLife.73097.sa2

---

## Additional files

### Supplementary files
• MDAR checklist

### Data availability
Links to data and code are included in the manuscript.

The following dataset was generated:

| Author(s) | Year | Dataset title | Dataset URL | Database and Identifier |
|---|---|---|---|---|
| Zhu SL, Lakshminarasimhan KJ, Arfaei N, Angelaki DE | 2022 | Gaze-navigation | https://gin.g-node.org/neuro-sci/gaze-navigation | G-node, neuro-sci/gaze-navigation |

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

# Appendix 1

Relevance simulations: Motivating the quantitative characterization of the task relevance of visual samples, we show an illustration of the consequence of mistaken beliefs about the passability of specific transitions on the subjective value function. Transition *toggling* can be defined as the act of removing an obstacle between two states if an obstacle was previously present, or adding an obstacle at that location if one was previously absent. In alignment with intuition, some transitions are more important to veridically represent (*Appendix 1—figure 1A* – middle), as toggling them results in a dramatic change to the value function (which is essential to computing the optimal set of actions to reach the goal), while toggling some other transitions causes a relatively minimal change (*Appendix 1—figure 1A* – right).

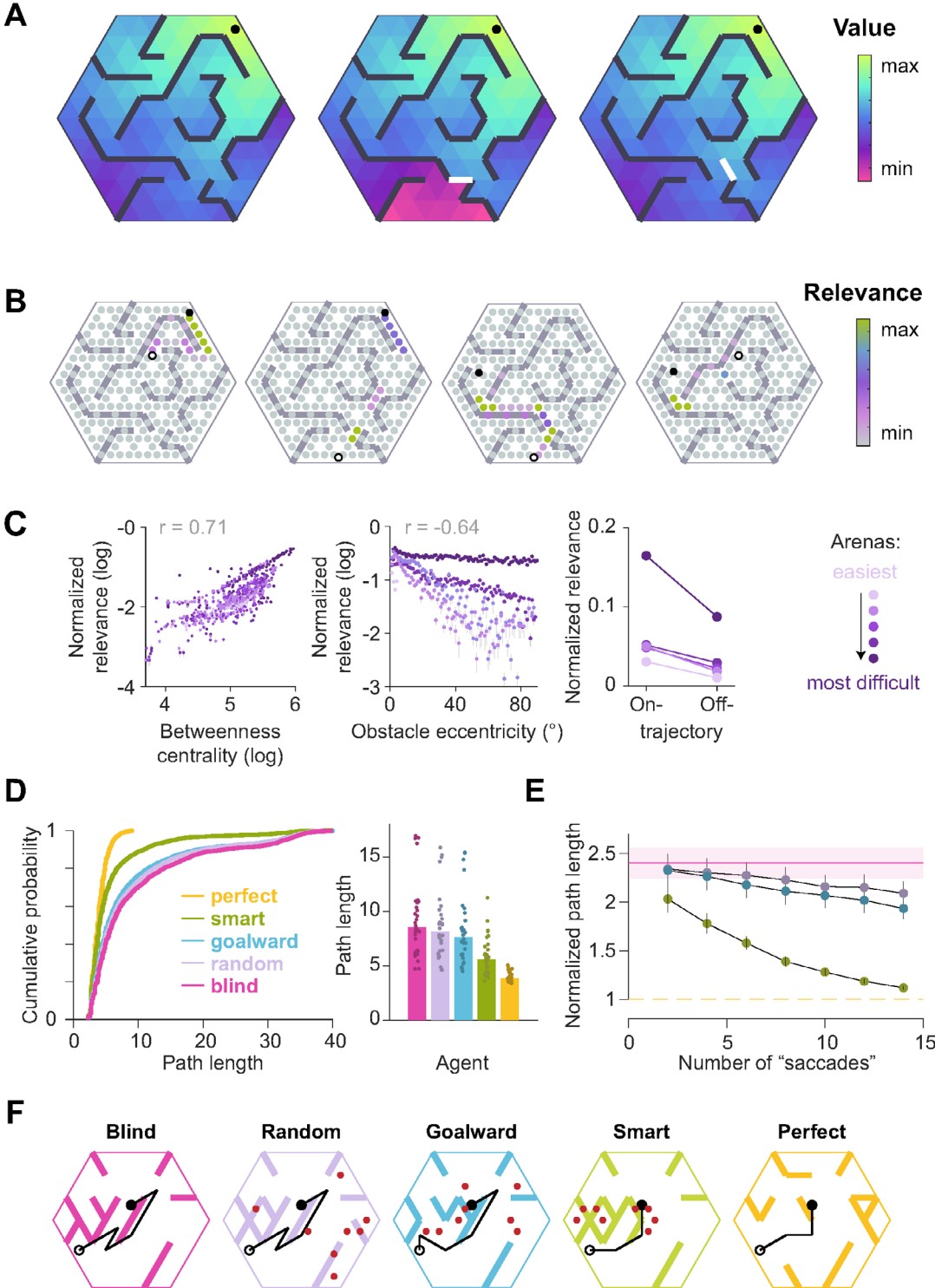

**Appendix 1—figure 1.** Simulations validate the utility of precisely knowing the status of theoretically important transitions. (**A**) Value functions corresponding to an arbitrary goal location (closed circle) in an example arena (left) and in the arenas resulting from blocking either a bottleneck transition (center) or a transition that was not a bottleneck (right). (**B**) Theoretical relevance of all transitions (circles) for the example arena for four different pairs of start (open black circle) and goal (closed black circle) states. (**C**) The betweenness centrality of a state describes the degree to which the state controls the traffic flowing through the area (see Methods). Left: Across all possible start and end locations, the mean normalized relevance of non-obstacle transitions (across all possible start and goal state pairings) was positively correlated with betweenness centrality values. Middle: The mean normalized

*Appendix 1—figure 1 continued on next page*

*Appendix 1—figure 1 continued*

relevance of obstacles was negatively correlated with the eccentricity of each obstacle from the straight line connecting the current state to the goal state. Right: Transitions that fell on the optimal trajectory had greater relevance than those that fell outside of it. (**D**) Simulation results of an agent instantiated with a perfect transition model (orange), and four agents with imperfect transition models, three of whom were endowed with the ability to correct their model according to different rules (see text). Those three agents were allowed to make eight 'saccades', each of which could update one transition. Left: Cumulative distributions of the path lengths of various agents (100 trials each from 25 different arenas; see Methods). Right: Median of trial-averaged path lengths across all simulated arenas; data points denote trial-averaged path lengths in individual arenas. (**E**) Results of simulations similar to D but with a variable budget of 'saccades'. Each line denotes the average path length (across arenas) of one agent as a function of the number of 'saccades'. For each trial, path lengths of different agents were normalized by the optimal path length before trial-averaging. Error bars denote ±1 SEM. (**F**) Example simulated trajectories, as well as the gaze samples (red dots, if applicable), taken by each agent. The configuration of the arena reflects the agent's subjective model at the *end* of all eye movements. Note that the subjective model of the 'smart' agent was still quite mismatched with the true world model after eight eye movements, but the visual samples allowed for the correction of the model at crucial locations such that the trajectory of the 'smart' agent was closer to optimal than that of the other agents.

By defining relevance of transitions according to *Equation 1*, we can thus capture multiple task-relevant attributes in a succinct manner. Theoretically investigating whether looking at task-relevant transitions improves navigational efficiency, we simulated artificial agents performing the same task that we imposed upon our human participants. One agent ('perfect') had a veridical subjective model of the environment, and thus was capable of computing the optimal trajectory (*Appendix 1—figure 1f*). Its antithesis ('blind') had an incorrect subjective model where half of the obstacle positions were 'misremembered' (toggled) to simulate a predicament where the agent had previous exposure to the arena, but were only halfway to learning the precise transition structure. The blind agent was incapable of using vision to correct their model prior to taking actions according to their subjectively computed value functions. Performance at these two extremes was compared against the performance of three agents that were allocated a fixed budget of 'saccades' to rectify their incorrect models. These agents either randomly interrogated transitions ('random'), preferentially sampled transitions along the direction connecting the agent's starting location to the goal location ('goalward'), or chose the most task-relevant transitions as defined by the relevance metric in *Equation 1* ('smart').

While all three agents showed an improvement over the 'blind' agent, the agent with knowledge about the most task-relevant transitions resulted in much shorter average path lengths than agents looking at transitions along the general direction of the goal or looking at random transitions (mean path length ±SE – perfect: 3.9 ± 0.1, blind: 9.6 ± 0.7, random: 8.9 ± 0.6, goalward: 8.6 ± 0.7, smart: 5.8 ± 0.3; *Appendix 1—figure 1D*). Moreover, the performance of the smart sampling agent quickly approached optimality as the number of sampled transitions increased (*Appendix 1—figure 1E*). The rate of performance improvement was substantially slower for the goalward and random samplers (linear rather than exponential). These results were robust to the precise algorithm used to compute the value function in *Equation 1*. In particular, the successor representation (SR) has been proposed as a computationally efficient, biologically plausible alternative to pure model-based algorithms like value iteration for responding to changing goal locations (*Momennejad et al., 2017*; *Stachenfeld et al., 2017*). We found that estimating the task-relevance of transitions using values implied by SR resulted in a similar performance improvement (*Appendix 1—figure 2*). Nevertheless, we emphasize that our objective was to use the relevance metric simply as a means to probe whether humans preferentially looked at task-relevant transitions. Understanding how the brain might compute such metrics is outside the scope of this study.

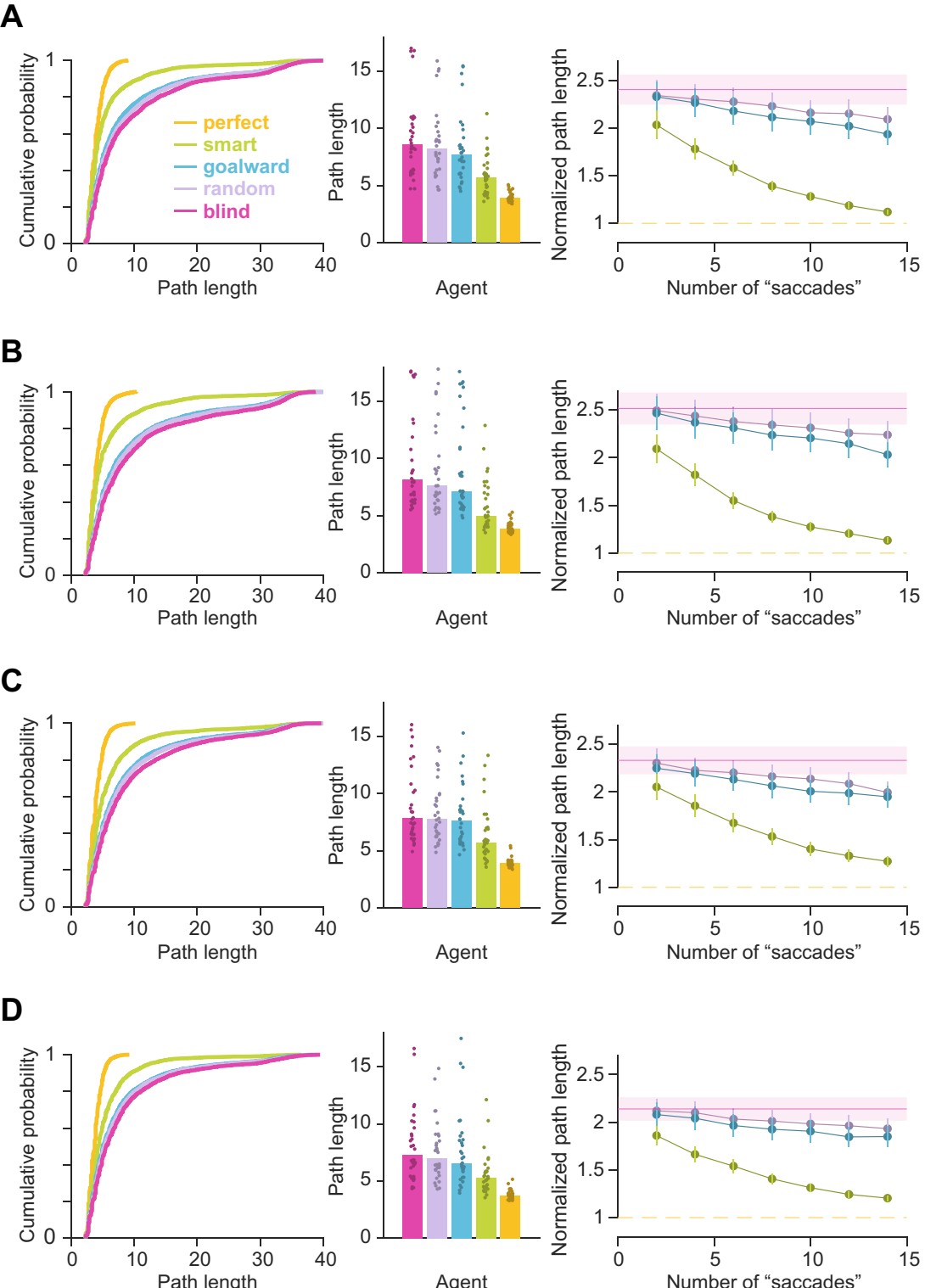

**Appendix 1—figure 2.** Simulations reveal that foveating 'relevant transitions' reduces path length. Results were robust to the precise algorithm (value iteration vs. successor representation) as well as the degree of temporal abstraction (current state vs optimal trajectory) used to estimate the relevance of transitions. Plots similar to *Figure 3C and D* are shown for relevance values calculated with **A** value iteration, current state, **B** value iteration, entire trajectory, **C** successor representation, current state, and **D** successor representation, entire trajectory.

Relevance derivation: In this section, we derive a general measure to quantify the relevance of transitions with respect to the task of navigating between two given states. The following derivation

focuses on the general setting when external noise (stochastic transitions) is present and internal noise (model uncertainty) is inhomogeneous. As we will show, the measure used to quantify transition relevance in the main text (*Equation 1*) corresponds to the special case where transitions are deterministic and uncertainty is homogeneous. Let $T_k$ denote the status of the $k^{th}$ stochastic transition (1 or 0) and $p_k$ be the parameter of the true probability distribution (p.d.) of that transition such that $P(T_k = 1) = p_k$ and $P(T_k = 0) = 1 - p_k$. Let $\hat{p}_k$ be the parameter of the subjective probability distribution of the transition. That is the agent thinks that $P(T_k = 1) = \hat{p}_k$ and $P(T_k = 0) = 1 - \hat{p}_k$. Given a particular goal state, let $V_s^k$ denote the value of the agent's current state $s$ evaluated using the true transition status $T_k$ such that $V_s^k = V_s(T_k = 1)$ if $T_k = 1$ and $V_s^k = V_s(T_k = 0)$ if $T_k = 0$. Let $\hat{V}_s^k$ denote the expectation of the value of state $s$ evaluated using the subjective transition p.d. of the $k^{th}$ transition such that $\hat{V}_s^k = \hat{p}_k gt V_s(T_k = 1) + (1 - \hat{p}_k) gt V_s(T_k = 0)$. Since looking at a transition will dramatically reduce the uncertainty about the status of that transition, this can impact the subjective value of the current state, provided that transition is critical to the task at hand. For instance, discovering a subway line linking your neighborhood and downtown will increase the value of your neighborhood if your workplace is located downtown, but will have no impact if your workplace is located crosstown. Therefore, we define relevance ($\Omega_k$) of the $k^{th}$ transition as the expectation of the (log) change in subjective value about the current state $s$ induced by looking at that transition. Then, we have:

$\Omega_k = \mathbb{E}[\log(|V_s^k - \hat{V}_s^k|)]_{p_k}$ where $\mathbb{E}[.]_{p_k}$ denotes expectation taken w.r.t. the true p.d.

$$= \mathbb{E}[\log(|V_s^k - \hat{p}_k V_s(T_k = 1) - (1 - \hat{p}_k)V_s(T_k = 0)|)]$$

$$= p_k \log(|V_s(T_k = 1) - \hat{p}_k V_s(T_k = 1) - (1 - \hat{p}_k)V_s(T_k = 0)|)+$$

$$(1 - p_k)\log(|V_s(T_k = 0) - \hat{p}_k V_s(T_k = 1) - (1 - \hat{p}_k)V_s(T_k = 0)|)$$

$$= p_k \log((1 - \hat{p}_k)|\Delta V|) + (1 - p_k)\log((\hat{p}_k)|\Delta V|) \text{ where } \Delta V = V_s(T_k = 1) - V_s(T_k = 0)$$

$$= p_k \log(1 - \hat{p}_k) + (1 - p_k)\log(\hat{p}_k) + \log(|\Delta V|)$$

$$= (p_k - 1 + 1)\log(1 - \hat{p}_k) - p_k \log(\hat{p}_k) + \log(\hat{p}_k) + \log(|\Delta V|)$$

$$= -(1 - p_k)\log(1 - \hat{p}_k) - p_k \log(\hat{p}_k) + \log(\hat{p}_k) + \log(1 - \hat{p}_k) + \log(|\Delta V|)$$

$$= H(p_k, \hat{p}_k) > + > \log(\hat{p}_k > (1 - \hat{p}_k)) > + > \log(|\Delta V|)$$

where $H(X, Y)$ denotes the cross entropy between $X$ and $Y$

$$= H(p_k) > + > D_{KL}(p_k \| \hat{p}_k) > + > \log(\hat{p}_k > (1 - \hat{p}_k)) > + > \log(|\Delta V|)$$

where $H(X)$ denotes the entropy of $X$

$$= H(p_k) > + > D_{KL}(p_k \| \hat{p}_k) > + > \log(\text{Var}[\hat{T}_k]) > + > \frac{1}{2}\log(|\Delta V|^2)$$

where $\hat{T}_k$ denotes the subjective knowledge about the status of the $k^{th}$ transition

Observe that $\Omega_k$ is comprised of four factors: (I) $H(p_k)$, the entropy of $p_k$, which captures transition volatility, (II) $D_{KL}(p_k \| \hat{p}_k)$, the Kullback-Leibler divergence between true and subjective p.d., which captures the degree of mismatch between the subjective and true transition models, (III) $\log(\text{Var}[\hat{T}_k])$, the log variance of the subjective status of the transition, which captures the agent's uncertainty, and (IV) $\log(|\Delta V|^2) = \log([V_s(T_k = 1) - V_s(T_k = 0)]^2)$, the log change in the value of the current state induced by changing the transition status, which captures the sensitivity of the value function to the transition. The first and third terms suggest that an agent should prioritize looking at transitions with high volatility and high subjective uncertainty. The second term suggests that it is best to look at transitions whose subjective status is known to be wrong. Although this is mathematically correct, agents would not know the true model to begin with and therefore cannot direct their attention at such transitions. Therefore, if external and internal noise are homogeneous, the best strategy would be to look at transitions which the value function is highly sensitive to, as postulated by *Equation 1* in the main text. Note that in deriving $\Omega_k$, we have neglected the contribution of model mismatches

that may exist at other transitions. This approximation will be valid if the model mismatch is small, and the solution works well in practice as demonstrated by the simulations (*Figure 1*).

# Appendix 2

**Appendix 2—table 1.** The order of arena presentation was randomized across participants.

| Participant ID | Block 1 | Block 2 | Block 3 | Block 4 | Block 5 |
|---|---|---|---|---|---|
| 1 | 3 | 2 | 1 | 4 | 5 |
| 2 | 5 | 4 | 1 | 2 | 3 |
| 3 | 5 | 3 | 1 | 2 | 4 |
| 4 | 4 | 5 | 2 | 3 | 1 |
| 5 | 3 | 4 | 2 | 5 | 1 |
| 6 | 5 | 2 | 1 | 3 | 4 |
| 7 | 3 | 2 | 5 | 1 | 4 |
| 8 | 4 | 5 | 3 | 1 | 2 |
| 9 | 2 | 1 | 4 | 5 | 3 |
| 10 | 5 | 3 | 1 | 2 | 4 |
| 11 | 2 | 3 | 4 | 5 | 1 |
| 12 | 1 | 2 | 5 | 3 | 4 |
| 13 | 3 | 2 | 5 | 1 | 4 |

**Appendix 2—table 2.** Median true relevance values (vs median shuffled relevance) for each arena ($\times 10^{-3}$).

Arenas 1–5 are in the order of least to greatest complexity.

| Epoch | Arena 1 | Arena 2 | Arena 3 | Arena 4 | Arena 5 |
|---|---|---|---|---|---|
| Search | 0 (vs 0) | 0 (vs 0) | 0 (vs 0) | 1.4 (vs 0.1) | 5.3 (vs 1.4) |
| Pre-movement | 32 (vs 0) | 31 (vs 0) | 45 (vs 0) | 47 (vs 0) | 137 (vs 6.2) |
| Movement | 0 (vs 0) | 9.5 (vs 0) | 47 (vs 0.7) | 51 (vs 4.3) | 201 (vs 60) |

**Appendix 2—table 3.** Number of participants (13 total) with a significant Pearson's correlation ($p \leq 0.05$) between the dependent variable and each independent variable (number of turns, path length, bearing angle, number of trajectory options).

Note: This correlation analyses does not characterize conditional dependencies, which may also be present in the data. Such dependencies are factored into the LME model described elsewhere.

| Figure(s) | Dependent variable | # Turns | Length | Bearing | # Options |
|---|---|---|---|---|---|
| 1i | Pre-movement epoch duration | 12 | 12 | 10 | 1 |
| 1i | Movement epoch duration | 13 | 13 | 12 | 5 |
| 2b | Variance of gaze pre-move. | 13 | 13 | 11 | 5 |
| 2b | Variance of gaze move. | 10 | 13 | 7 | 6 |
| 2d | Gaze @ goal duration pre-move. | 13 | 13 | 10 | 8 |
| 2d | Gaze @ goal duration move. | 13 | 13 | 12 | 10 |
| 2 f | Gaze distance from goal pre-move. | 13 | 13 | 11 | 7 |
| 2 f | Gaze distance from goal move. | 13 | 13 | 11 | 7 |
| 4 c, 4d | % Time sweeping backward pre-move. | 10 | 8 | 7 | 2 |
| 4 c, 4d | % Time sweeping backward move. | 12 | 12 | 4 | 3 |
| 4 c, 4e | % Time sweeping forward pre-move. | 2 | 3 | 3 | 2 |
| 4 c, 4e | % Time sweeping forward move. | 13 | 12 | 12 | 4 |

**Appendix 2—table 4.** Number of participants (13 total) with a significant Pearson's correlation (p≤0.05) between the dependent variable and arena complexity (for analyses which require pooling trials).

A linear mixed effects model (LME) with random slopes and intercepts yielded participant-specific slopes, from which we computed the mean and coefficient of variation (CV). Note that results showed low between-participant variability. All variables were z-scored prior to model fitting.

| Figure(s) | Dependent variable | Mean slope | CV slope |
|---|---|---|---|
| 2d | Gaze @ goal duration move. by # turns remaining | 0.80 | $5.6e^{-15}$ |
| 2 f | Gaze distance from goal move. by # turns remaining | −0.86 | $−2.2e^{-3}$ |

