## [Editor Report]

This beautiful piece of work demonstrates the power of eye movement analysis in understanding the cognitive algorithms for navigation, and more generally, for real-time planning and decision making. Its sophisticated computational measures of the multiple dimensions of eye movement data can potentially inspire discoveries in many fields of cognitive neuroscience concerning rich human behavior.

---

## [Decision Letter]

**Decision letter after peer review:**

Thank you for submitting your article "Eye movements reveal spatiotemporal dynamics of active sensing and planning in navigation" for consideration by *eLife*. Your article has been reviewed by 3 peer reviewers, and the evaluation has been overseen by a Reviewing Editor and Joshua Gold as the Senior Editor. The following individual involved in review of your submission has agreed to reveal their identity: Hugo J Spiers (Reviewer #2).

Essential revisions:

Below is a summary of what we think are most important for the authors to address if you choose to submit a revised manuscript.

1) Most of the conclusions reached by the authors seem to rely on fixed-effect analysis (i.e., with all participants pooled as one), which basically neglects between-participant variability and may thus inflate the Type-I error in statistics. May the same conclusions be reached when the random effects of participants are appropriately considered (e.g., using linear mixed-effects model analysis with random intercept and slopes)?

2) A related concern is the relatively small sample size compared to contemporary navigation studies, which should be justified in the paper. The current sample size would be acceptable if the major conclusions hold on the group level when random participant effects are appropriately considered or hold on the individual level for at least 8 of all 9 participants. Otherwise, the authors are recommended to increase the sample size of the study.

3) A few factors that covary with the complexity of the maze may influence eye movement and error patterns, which include the length of the optimal path, the number of alternative plausible paths, and the affordances linked to turning biases and momentum. These factors should receive some treatments in data analysis or at least in discussion. Please see Reviewer #2's comments for details.

4) Alternative theoretical accounts for participants' eye movement patterns, other than active sensing, should be considered.

*Reviewer #2 (Recommendations for the authors):*

It is worth being careful defining preplay and replay and pre-activation of activity. The pre-play of Dragoi and Tonegawa includes pre-play for visible places but also de novo preplay where rats have never seen the environment. There is distinct difference between non-local replay during sharp-wave-ripples (SWRs) and the theta oscillation state during running. It is worth making these patterns of hippocampal dynamics clearer for the readers. The findings of Johnson and Redish occur during the theta state, whereas the replay or pre-play of distant locations occur during SWRs.

What wasn't obvious what the advantage of using dynamic programming was over other methods to determine the optimal path.

I didn't see any discussion of how this approach could be applied across species. It seems a great strategy to explore planning in non-human primates and I assume this is the plan given the expertise in the research group.

It may be useful to cite this work as it is a real-world example, but lacks the computational approach taken by the authors: Ghamari, H., and Golshany, N. (2021). Wandering Eyes: Using Gaze-Tracking Method to Capture Eye Fixations in Unfamiliar Healthcare Environments. HERD: Health Environments Research and Design Journal, 19375867211042344.

---

## [Author Response]

Essential revisions:Below is a summary of what we think are most important for the authors to address if you choose to submit a revised manuscript.1) Most of the conclusions reached by the authors seem to rely on fixed-effect analysis (i.e., with all participants pooled as one), which basically neglects between-participant variability and may thus inflate the Type-I error in statistics. May the same conclusions be reached when the random effects of participants are appropriately considered (e.g., using linear mixed-effects model analysis with random intercept and slopes)?

The revised manuscript includes new results from fitting linear mixed-effects models which consider the participant-specific random effects. We find that all major trends are shared across participants even after accounting for variability across participants.

2) A related concern is the relatively small sample size compared to contemporary navigation studies, which should be justified in the paper. The current sample size would be acceptable if the major conclusions hold on the group level when random participant effects are appropriately considered or hold on the individual level for at least 8 of all 9 participants. Otherwise, the authors are recommended to increase the sample size of the study.

We conducted a new round of experiments by recruiting more participants, increasing our sample size to 13. We also share data in the tables in Appendix 2 showing that main results also hold on the individual level for a vast majority of the participants.

3) A few factors that covary with the complexity of the maze may influence eye movement and error patterns, which include the length of the optimal path, the number of alternative plausible paths, and the affordances linked to turning biases and momentum. These factors should receive some treatments in data analysis or at least in discussion. Please see Reviewer #2's comments for details.

Since path length and turns covary with maze complexity, they have now been added as covariates in a linear mixed effects model to predict trial-by-trial variability in error patterns, pre-movement duration, as well as spatial and temporal features of eye movements. These results are presented in the revised manuscript (bar graphs in Figures 2, 4 Figure 1 —figure supplement 1, Figure 1 —figure supplement 2, Figure 4 —figure supplements 1, 2, and 3). We found that the number of alternative paths does not covary with complexity – both the simplest and the most complex maze have no non-trivial alternative trajectories. Therefore, we separately analyzed the effect of the number of alternative paths on eye movements both across time within a trial (Figure 5D) and across trials (Figure 4 —figure supplement 1D).

4) Alternative theoretical accounts for participants' eye movement patterns, other than active sensing, should be considered.

In the revised manuscript, we reinterpret our results in the light of the alternative account and propose testable predictions for future experiments. To achieve a more balanced treatment, we also deemphasize the dichotomy between active sensing and planning throughout the revised manuscript, starting with the title.

Reviewer #2 (Recommendations for the authors):It is worth being careful defining preplay and replay and pre-activation of activity. The pre-play of Dragoi and Tonegawa includes pre-play for visible places but also de novo preplay where rats have never seen the environment. There is distinct difference between non-local replay during sharp-wave-ripples (SWRs) and the theta oscillation state during running. It is worth making these patterns of hippocampal dynamics clearer for the readers. The findings of Johnson and Redish occur during the theta state, whereas the replay or pre-play of distant locations occur during SWRs.

We agree that there are major differences between the two major types of sequential activity observed in the medial temporal lobe. As we currently do not have enough evidence to speculate about whether gaze sweeps are similar to SWRs or theta-timed activation, we refrained from citing any literature regarding the theta rhythm and avoided discussing replay in detail, but we plan to investigate the similarities in greater detail in future.

What wasn't obvious what the advantage of using dynamic programming was over other methods to determine the optimal path.

We tried to improve the clarity of the sentence starting with “the optimal value function can be computed... ” by specifying that dynamic programming is a very efficient method for such a computation. Concretely, this technique returns the value function, which can be reused to compute the optimal path from any location to that particular goal location by ascending the value function without re-running the algorithm. (The computation times were manageable for medium-sized state spaces like the arenas we used, but this algorithm can become tedious for more complex tasks, and in this case, we would use other algorithms which approximate the optimal solution.)

I didn't see any discussion of how this approach could be applied across species. It seems a great strategy to explore planning in non-human primates and I assume this is the plan given the expertise in the research group.

Your assumption is correct. In the third and fourth paragraphs of the Discussion (line 407 and 431), we added that we would be interested in studying neural information flow between brain areas in different species, and elaborated upon which brain regions we would be interested in recording from and why.

“Neurally, this could be implemented by circuits that exert executive control over voluntary eye movements. Candidate substrates include the dorsolateral prefrontal cortex, which is known to be important for contextual information processing and memory-guided saccades […], and the anterior cingulate cortex, which is known to be involved in evaluating alternative strategies […]. To better understand the precise neural mechanisms underlying the spatial gaze patterns we observed, it would be instructive to examine the direction of information flow between the oculomotor circuitry and brain regions with strong spatial and value representations during this task in animal models.

In light of the similarities between sweeps and sequential hippocampal activations, we predict that direct or indirect hippocampal projections to higher oculomotor controllers (e.g. the supplemental eye fields through the orbitofrontal cortex) may allow eye movements to embody the underlying activations of state representations… the contents of gaze have been found to influence activity in the hippocampus […] and entorhinal cortex […]. Therefore, it is conceivable that sequential neural activity could emerge from consolidating temporally extended eye movements such as sweeps. We hope that in future, simultaneous recordings from brain areas involved in visual processing, eye movement control, and the hippocampal formation would uncover the mechanisms underlying trajectory sweeping eye movements and their relationship to perception and memory.”

It may be useful to cite this work as it is a real-world example, but lacks the computational approach taken by the authors: Ghamari, H., and Golshany, N. (2021). Wandering Eyes: Using Gaze-Tracking Method to Capture Eye Fixations in Unfamiliar Healthcare Environments. HERD: Health Environments Research and Design Journal, 19375867211042344.

Thank you for the reference. This is cited in the discussion as an example of a study that only allows a restricted field of view.